# Young adult born neurons enhance hippocampal dependent performance via influences on bilateral networks

Jia-Min Zhuo[1]*, Hua-an Tseng[1†], Mitul Desai[2†], Mark E Bucklin[1], Ali I Mohammed[1], Nick TM Robinson[3], Edward S Boyden[2,4], Lara M Rangel[3], Alan P Jasanoff[2], Howard J Gritton[1], Xue Han[1]*

[1]Biomedical Engineering Department, Boston University, Boston, United States; [2]Department of Bioengineering, McGovern Institute, Cambridge, United States; [3]Department of Psychology, Boston University, Boston, United States; [4]Media Lab, Massachusetts Institute of Technology, Cambridge, United States

**Abstract** Adult neurogenesis supports performance in many hippocampal dependent tasks. Considering the small number of adult-born neurons generated at any given time, it is surprising that this sparse population of cells can substantially influence behavior. Recent studies have demonstrated that heightened excitability and plasticity may be critical for the contribution of young adult-born cells for certain tasks. What is not well understood is how these unique biophysical and synaptic properties may translate to networks that support behavioral function. Here we employed a location discrimination task in mice while using optogenetics to transiently silence adult-born neurons at different ages. We discovered that adult-born neurons promote location discrimination during early stages of development but only if they undergo maturation during task acquisition. Silencing of young adult-born neurons also produced changes extending to the contralateral hippocampus, detectable by both electrophysiology and fMRI measurements, suggesting young neurons may modulate location discrimination through influences on bilateral hippocampal networks.

*For correspondence:
jmzhuo2002@gmail.com (J-MZ);
xuehan@bu.edu (XH)

†These authors contributed equally to this work

**Competing interests:** The authors declare that no competing interests exist.

## Introduction

Accumulating evidence has revealed that neurogenesis continues in the mammalian hippocampus, including that of humans throughout life (*Déry et al., 2013*; *Spalding et al., 2013*). Neurons generated in the dentate gyrus (DG) area of the hippocampus during adulthood mature into dentate granule cells (DGCs), which can integrate into existing hippocampal neural circuits (*Toni et al., 2008*; *Gu et al., 2012*) and influence affective states or cognitive processes (*Santarelli et al., 2003*; *Snyder et al., 2011*). In particular, recent studies applying pharmacogenetic and optogenetic tools to inhibit adult neurogenesis have demonstrated how this population of new cells can profoundly influence performance in certain hippocampal dependent tasks, such as those that assess spatial pattern separation and cognitive flexibility (*Clelland et al., 2009*; *Gu et al., 2012*; *Swan et al., 2014*).

While adult born dentate granule cells (abDGCs) continue their morphological development for an extended period of time, often over months in rodents and years in primates (*Ge et al., 2007*; *Kohler et al., 2011*), there is a transient maturation period corresponding to the first several weeks of integration when they exhibit heightened membrane excitability and elevated potential for synaptic plasticity (*Ge et al., 2007*; *Marín-Burgin et al., 2012*; *Dieni et al., 2013*; *Kropff et al., 2015*). The biophysical properties of young abDGCs, during their highly excitable critical developmental window may be uniquely suited to facilitate the computational aspects that underlie pattern

separation by influencing larger networks (*Sahay et al., 2011a, 2011b*; *Yassa and Stark, 2011*). It is plausible, for example, that young abDGCs may be easily excited by entorhinal inputs that recruit distinct neuronal ensembles, and thus constitute a unique pattern of activity required for pattern separation (*Ge et al., 2007*; *Sahay et al., 2011b*). Alternatively, abDGCs with presumed higher baseline firing rates may contribute to pattern separation by non-selectively driving global feedback inhibition via hilar interneurons, thereby creating a sparse firing state within the DG, in which only the strongest or most highly trained inputs can overcome local inhibition (*Jung and McNaughton, 1993*; *Alme et al., 2010*; *Neunuebel and Knierim, 2012*; *Ikrar et al., 2013*; *Piatti et al., 2013*).

Although the mechanism(s) by which this process occurs have remained elusive, it is becoming increasingly clear that this young maturing population of abDGCs, despite their small numbers, have the potential to robustly influence behavior. In order to better understand how these cells integrate into behavioral networks and influence hippocampal function, we performed a series of experiments using a location discrimination (LD) task designed to test spatial pattern separation, an ability shown to be critically dependent on hippocampal neurogenesis. Pattern separation refers to a computational process of forming distinct representations of similar inputs via unique ensembles of neurons, which could facilitate memory encoding by minimizing interference (*Sahay et al., 2011b*; *Yassa and Stark, 2011*). It has been hypothesized that the DG can perform this function as a consequence of the sparse firing properties of its cells and the large number of dentate cells compared to both its upstream and downstream regions (*Treves and Rolls, 1994*; *Leutgeb et al., 2007*; *Yassa and Stark, 2011*). While pattern separation could facilitate the ability to distinguish between experiences that are similar, it has been difficult to test how the adult-born DGCs may contribute to this ability. A few behavioral studies have provided evidence that the DG, and the existence of abDGCs in particular, are relevant for discriminating between similar experiences or spatial locations (*Clelland et al., 2009*; *Yassa and Stark, 2011*; *Déry et al., 2013*). For example, selective lesions of the DG but not CA1 or CA3 impaired rats' ability to discriminate between locations in spatial discrimination tasks (*Treves and Rolls, 1994*; *Leutgeb et al., 2007*; *Yassa and Stark, 2011*). Studies in humans have also shown that increases in blood-oxygen-level dependent (BOLD) signal in the DG is highly correlated with the pattern separation process (*Bakker et al., 2008*). Interestingly, even though the impact of adult neurogenesis on other hippocampal-dependent behaviors remains controversial (*Clelland et al., 2009*; *Deng et al., 2010*; *Sahay et al., 2011a*; *Kesner, 2013*), chronic increases or decreases of adult neurogenesis have provided consistent results in behavioral tasks designed to test abilities that are thought to depend on pattern separation (*Clelland et al., 2009*; *Sahay et al., 2011a*; *Kesner et al., 2014*; *Rangel et al., 2014*).

Using a combination of optogenetics, fMRI, electrophysiology, and a behavioral task designed to assess pattern separation, we discovered that abDGCs are fundamental to pattern separation performance and recruited into behaviorally relevant networks through the process of learning. We also discovered that silencing young adult-born cells leads to enhanced hippocampal excitability and that this population has a specialized role in influencing hippocampal function through the regulation of bilateral hippocampal neural networks.

## Results

### Adult neurogenesis contributes to location discrimination (LD) performance

Retroviruses are known to specifically integrate into newly divided cells making them an optimal technique for targeting neurogenesis. In the central nervous system, the vast majority of newly divided cells in the subgranular zone will go on to become granule cells (*Kuhn et al., 1996*). Therefore, we designed a Murine Stem Cell Virus (MSCV) retrovirus (*Hawley et al., 1994*) with a *synapsin* promoter (SYN1) to drive the selective expression of the optogenetic silencer ArchT (*Han et al., 2011*) only in newborn cells that will take on a neuronal fate (*Figure 1a*). However, to verify and quantify the time course and specificity of MSCV targeting, we also characterized the co-localization of ArchT-GFP or GFP reporter labeling with the non-specific dividing cell marker EdU injected at 0, 3 and 6 days after retrovirus injection. EdU was delivered systemically, resulting in the labeling of all newly dividing cells in both hemispheres across the dorsal-ventral axis of the hippocampus. Although, MSCV was delivered intracranially in one injection per hemisphere, we found that a single

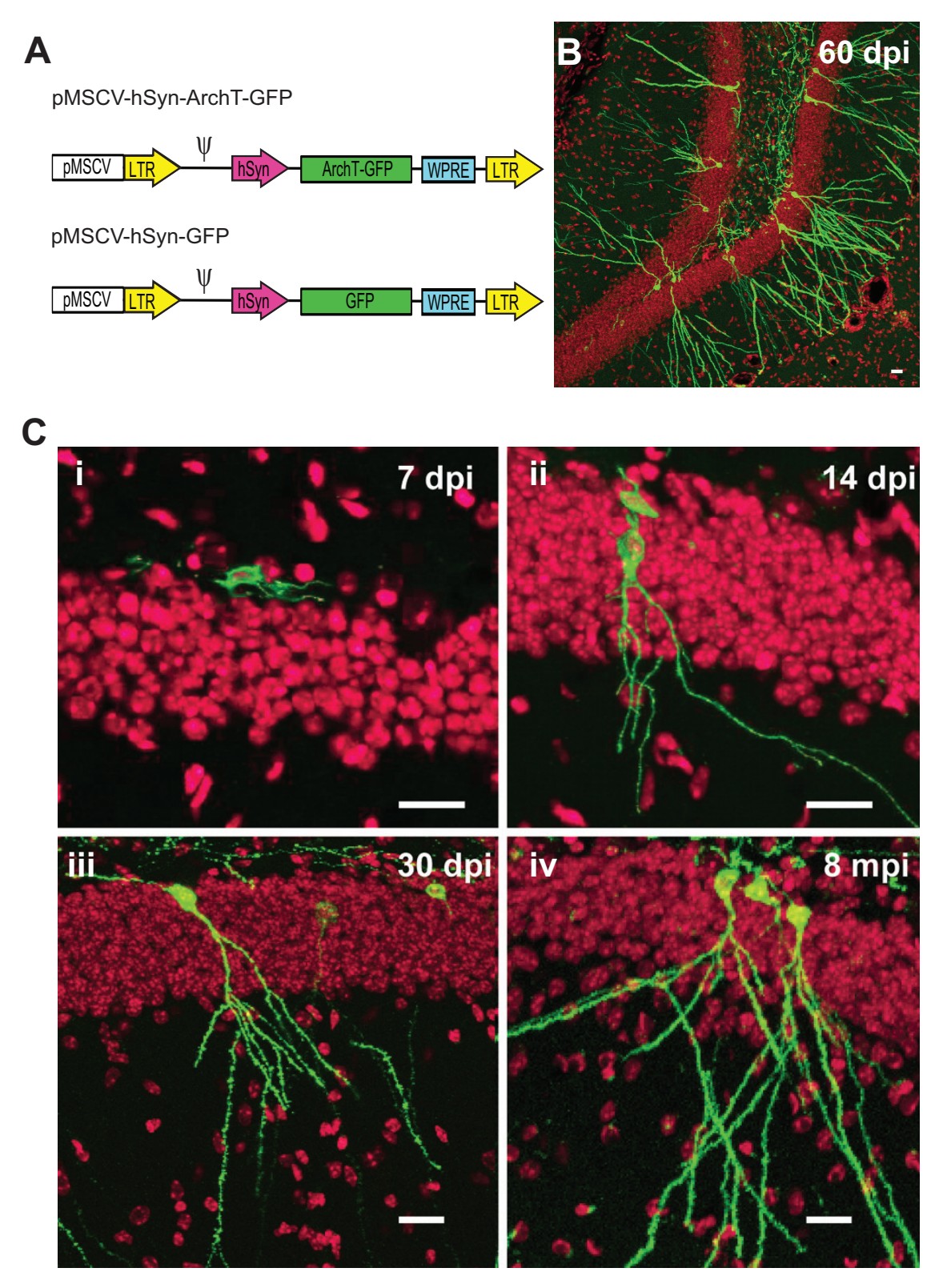

**Figure 1.** Selective labeling of age-defined abDGCs with the optogenetic silencer ArchT using retrovirus. (**A**) The schematics of the retroviral vectors (LTR: long terminal repeats; ψ: psi virus packaging sequence). (**B**) ArchT-GFP fluorescence in labeled abDGCs in the DG of an adult mouse, examined at 60 days post-injection (dpi). (**C**) ArchT-GFP fluorescence in labeled abDGCs examined at (**i**) seven dpi, (**ii**) 14 dpi, (**iii**) 30 dpi, and (**iv**) eight months post-injection. (Scale bar, 20 μm; green: ArchT-GFP fluorescence; red: TO-PRO-3 nuclei stain.).

injection preferentially labeled cells that would begin differentiating into abDGC over the first few days following virus injection. We found that 2.5 ± 1.5% of all EDU-labeled cells co-localized with the GFP transgene delivered via MSCV in animals given EdU injections three days post-viral injection. This number fell to 0.5 ± 0.3% of all EdU-labeled cells for animals given EdU injections on day seven post-virus (n = 3 mice for each group). This finding also validates that our MSCV retrovirus construct drove expression of ArchT-GFP uniquely in cells that were undergoing division during the short period following viral delivery, consistent with other retroviral studies (*Faulkner et al., 2008*; *Toni et al., 2008*). Among GFP-expressing cells examined two weeks after viral injections, almost all exhibited clear neuronal morphology, consistent with the preferred neuronal targeting properties of the *synapsin* promoter (*Nathanson et al., 2009*). Labeled abDGCs imaged 30–60 days post injection showed typical dentate cell morphology (*Figure 1B, C*) (*Zhao et al., 2006*), and ArchT-GFP expression was well targeted to the plasma membrane and remained strong in animals tested as far out as eight months post-viral injection (*Figure 1C*).

To examine the contribution of adult neurogenesis to the pattern separation ability of the DG, we tested the involvement of abDGCs in influencing the behavioral performance of a location discrimination (LD) task (*Clelland et al., 2009*; *Kesner, 2013*). This task was designed to quantitatively measure the animal's ability to discriminate between two objects separated spatially from one another across various distances (*Figure 2A*), and has been shown to be sensitive to hippocampal lesions (*McTighe et al., 2009*) or chronic ablation of adult neurogenesis using x-ray irradiation (*Clelland et al., 2009*). Specifically, we trained mice to separate two illuminated windows from one another in order to receive a water reward. The reward was obtained by returning to the same illuminated window on a given set of trials, defined as a block. Within a daily behavioral session, the same two windows remained illuminated but the rewarded window switched once an animal performed seven out of eight consecutive trials correctly. The following trial constituted the first trial of the subsequent block. Animals would continue for a total of 60 trials with the rewarded window alternating between the two illuminated ports as each block was successfully completed. Task difficulty was manipulated by varying the spatial distance parametrically between the two illuminated windows, separated by either one dark window (the difficult separation condition), three dark windows in between the illuminated windows (the medium separation condition), or five dark windows in between the illuminated windows (the easy separation condition). Further, the medium separation condition was only used for task training, and silencing sessions were only conducted in the difficult and easy separation condition to reduce the total number of silencing sessions animals were exposed to across days. Each day, mice were trained or tested using only one separation condition. Trials were separated by a 10 s inter-trial interval (ITI). This task was used because it is well suited to studying pattern separation in tethered animals and has been extensively validated in x-ray irradiated animals previously (*Clelland et al., 2009*; *Creer et al., 2010*).

To evaluate the function of age-defined abDGCs in task performance, we injected retrovirus into the adult hippocampus, encoding for either the MSCV-ArchT-GFP virus (ArchT-GFP group, labeled 328 ± 24 neurons/DG, n = 7 mice) or the control MSCV-GFP virus (GFP-control group, labeled 327 ± 78 neurons/DG, n = 5 mice). By using MSCV for targeting, we could limit ArchT expression to a small window of time following infusion thereby allowing us to test the effects of silencing of GC populations at a specific age of development. In order to ensure high levels of expression and broad optogenetic influence, the virus was injected at four sites in the bilateral DG, and a fiber array containing four fibers was implanted to target each viral injection site. Continuous laser light was delivered upon trial initiation, and terminated when the animal selected an illuminated window (avg. duration 4.02 ± 0.24 s; n = 471 laser silencing trials in 10 mice, mean ± S.E.M). Laser illumination was delivered in the second and fourth blocks during the optogenetic testing days (illumination blocks), but not in the first or third block (baseline blocks) (*Figure 2A*), and only after a mouse performed two correct trails within three consecutive trials. Thus, laser light was delivered on 11.8 ± 0.9 trials of the 60 total trials on testing days (mean ± S.E.M., n = 40 testing days in 10 mice), and each mouse was optogenetically tested for two days under each condition to discourage the use of alternative strategies within the behavioral session.

In the baseline blocks without laser illumination, both ArchT-GFP and GFP-control groups performed equally well and required similar numbers of trials to complete a block in all conditions (*Figure 2C*). In addition, consistent with previous studies (*Clelland et al., 2009*; *Oomen et al., 2013*), mice needed fewer trials to complete the block in the easy separation condition relative to

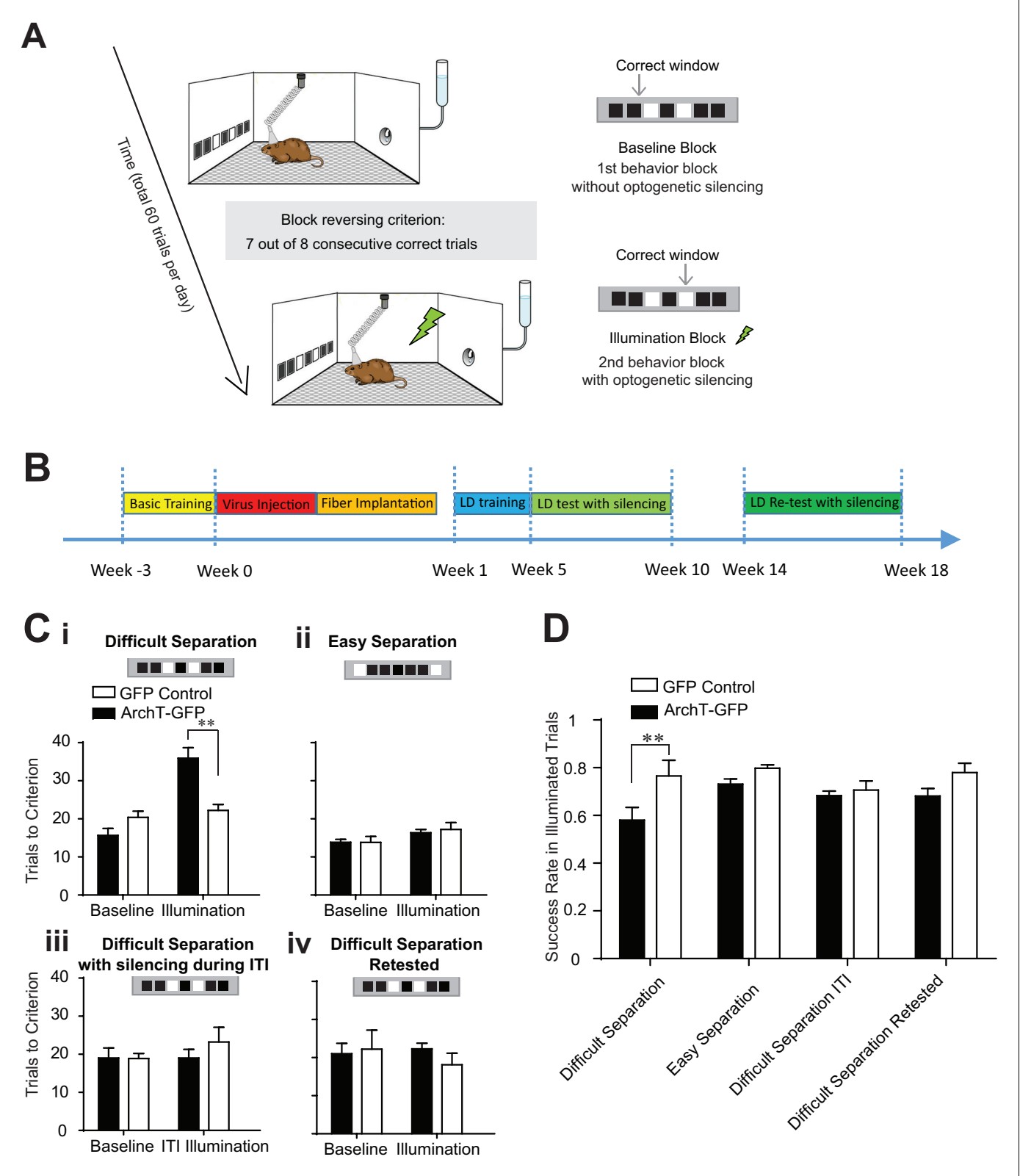

**Figure 2.** Transient optogenetic silencing of young abDGCs impaired behavioral performance and online information processing during location discrimination (LD). (A) Illustration of the LD task. The rewarded window was reversed when a mouse reached criterion performance in a given block. (B) Experimental timeline. (C) Behavioral performance measured as the number of trials to reach criterion in the baseline blocks and the laser illumination blocks, in the ArchT-GFP group and the GFP-control group, performing the difficult separation condition (i): [repeated measures two-way ANOVA,

*Figure 2 continued on next page*

*Figure 2 continued*

significant illumination effect: $F_{1,10}$ = 18.76, p<0.01; significant group effect $F_{1,10}$ = 6.28, p<0.05; significant mice group × illumination interaction: $F_{1,10}$ = 13.14, p<0.01; Post hoc Bonferroni tests: ArchT-GFP vs. GFP-control groups for illumination blocks: p<0.01; for baseline blocks: p>0.05], (ii) the easy separation condition, (iii) the difficult separation condition with optogenetic silencing during ITI, and (iv) the difficult separation condition during retest. Only the ArchT-GFP group's performance was impaired coincident with optogenetic silencing, and only in the difficult separation condition during the 5–10 week window. (D) The success rate achieved by various experimental conditions, calculated for trials where laser illumination was delivered. The success rate was significantly decreased in the ArchT-GFP group performing the difficult separation compared to the control group. [Repeated measure two-way ANOVA, significant group effect: $F_{1,8}$ = 11.50, p<0.01; Post hoc Bonferroni tests: ArchT-GFP vs. GFP-control groups for initial difficult separation: p<0.01; for the re-test: p>0.05]. **p<0.01. Error bars indicate.

the difficult separation condition (trials needed in easy separation: 16.0 ± 0.4; in difficult separation: 20.7 ± 1.0, p<0.01, n = 10 mice, paired t-test comparing the baseline blocks in easy versus difficult separation conditions). On trials where we delivered light to young abDGCs (5–10 weeks of maturation), the ArchT-GFP group showed selective impairment in performing the difficult separation, requiring more trials to complete the illumination blocks, but performed equally well as control mice under the easy separation condition (*Figure 2Ci*). Additionally, we found that some ArchT-GFP mice were incapable of reaching criterion during the illumination block under the difficult condition. We found no differences in reaction time, defined as time from trial initiation to window selection, across the two groups suggesting that optogenetic silencing of abDGCs is important for modulating the cognitive aspects of the task directly as opposed to influencing locomotion for example. These results also provide strong evidence that young abDGCs play an essential role in discriminating narrowly separated spatial locations, consistent with the theory of the DG's described role as a pattern separator.

To examine whether young abDGCs are selectively involved during the performing phase of the task, or have a more general role in task engagement, we transiently silenced the same abDGCs during the last 5 s of the ITI, and found no effect on behavioral performance in those blocks (*Figure 2Ciii*). This result demonstrates that young abDGCs are specifically required during the choice phase of the task. To further evaluate the impact of abDGCs on task performance during a given trial, we examined the success rate in trials when light was delivered. We found that transient silencing of young abDGCs also produced a significant reduction in single trial performance (*Figure 2D*). Together, these results suggest that adult neurogenesis is critical for online information processing during spatial discrimination.

Given that abDGCs maturation likely continues gradually over an extended period, and that these neurons show different biophysical properties at later stages of development, we retested the same mice 14–18 weeks after virus injection. This allowed us to track the extent of their behavioral influence longitudinally within the same mice (*Figure 2B*). Optogenetic silencing of older abDGCs (14–18 weeks of maturation) had no impact on behavioral performance in any of the separation conditions (*Figure 2Civ*). The baseline performance during retest was similar to that during the initial test, confirming stable behavioral performance over time and a maintained representation for task rules in these mice (*Figure 2C*). To determine if the inability of older cells to influence behavioral performance could be a result of neuronal loss, we quantified the number of labeled abDGCs at these early and late phases following virus injection. We found no significant loss of neurons in this group of animals relative to performing animals that underwent quantification at 10 weeks post-infusion

**Table 1.** ArchT-GFP cell label counts for abDGCs across different experimental groups.

| Group | GFP positive neurons per DG |
| --- | --- |
| Mice analyzed at 10 weeks post ArchT-GFP retroviral injection (n = 5) | 334 ± 43 |
| Longitudinal mice group analyzed at 18 weeks post ArchT-GFP retroviral injection (n = 5) | 297 ± 27 |
| Naïve mice group analyzed at 10 weeks post ArchT-GFP retroviral injection (n = 5) | 264 ± 40 |

(*Table 1*). This finding is consistent with the observation that although a significant number of abDGCs degenerate within the first couple weeks of birth, little degeneration occurs beyond 4 weeks of age (*Dayer et al., 2003*). Although unlikely, we cannot rule out the possibility that a small and histologically undetectable fraction of ArchT-GFP labelled neurons may degenerate given the variability among mice, which may be critical in influencing an individual animal's behavior. Whether the observed loss of behavioral impact during retest could be due to a slight reduction in cell number, or changes in the intrinsic excitability or synaptic targeting properties of the same number of cells; our results provide direct longitudinal evidence that a set of age-defined abDGCs that are highly effective in modulating LD performance when young, have comparatively weaker effects on performance at later developmental stages. This is consistent with the general observation that abDGCs exhibit differing levels of influence on behavior across maturation (*Gu et al., 2012*; *Nakashiba et al., 2012*; *Swan et al., 2014*).

To further determine if the DG remains relevant to the LD test during the retest phase, and that the process of memory consolidation did not eliminate the need for the DG's involvement (*Murray and Bussey, 2001*) in task performance, we performed similar behavioral analysis in POMC-Arch transgenic mice (*Figure 3A*). POMC-Arch mice express the optogenetic silencer Arch in all dentate granule cells, including both developmentally-born and adult-born cells, and the expression of ArchT in each cell persists permanently throughout the animal's life. (*McHugh et al., 2007*; *Madisen et al., 2012*). Silencing therefore reflects the effects of inactivating every granule cell, of every age simultaneously.

Optogenetic silencing of the DG in POMC-Arch mice were performed in an identical fashion as that in retrovirus injected mice groups described in *Figure 2*. We discovered that optogenetic silencing of the DG impaired the animal's ability to perform the difficult separation, during both the initial test and the retest phases, requiring more trials to complete the illumination block (*Figure 3B*), and exhibiting lower success rates (*Figure 3C*). In addition, optogenetic silencing in POMC-Arch mice did not produce a behavioral deficit during the easy separation condition (*Figure 3B*), confirming that the DG is required only for separating spatial locations that are finely delineated, but not for locations that are broadly dissociated, consistent with the general observation of the importance of the DG in behavioral tasks testing pattern separation (*Kesner, 2013*) (*Figure 3B*). As silencing in POMC-Arch mice represents complete inhibition of all DGCs simultaneously (including both young and mature neurons), our findings suggest that the DG has a critical and ongoing role in spatial discrimination during both the initial test and retest, and suggest that new young cells continue to support performance throughout the animal's lifetime. Further, our findings are supportive of the theory that the reduced behavioral influence of older abDGCs during retest is unlikely attributed to processes of memory consolation.

## Adult neurogenesis influences bilateral hippocampal regions during early development

To better understand the neural network mechanisms relevant for the observed behavioral impact of silencing young abDGCs during task performance, we performed fMRI in adult mice while optogenetically silencing abDGCs at different ages. MSCV-ArchT-GFP or control MSCV-GFP retrovirus was injected into the right DG at one site, labeling ~200 cells per DG (208 ± 20 neurons/DG for ArchT-GFP group, n = 8 mice, and 212 ± 35 neurons/DG for control group, n = 7 mice). Laser light was directed to the viral injection site through a surgically implanted optical fiber. Mice were lightly anesthetized, head-fixed, and scanned in a 9.4T scanner (*Figure 4A*). BOLD signals with a voxel size of 156 µm × 156 µm × 750 µm were acquired using a gradient-echo planar imaging protocol, and mapped to the high-resolution T2 weighted anatomical images acquired at a voxel size of 58 µm × 58 µm × 750 µm. Using fMRI, we were able to detect BOLD signal changes across the whole brain, while silencing abDGCs at different maturation stages in the same animals longitudinally. We performed the first fMRI analysis while optogenetically silencing abDGCs at seven weeks post viral injection, matching the initial behavioral test time-point, and then again at 16 weeks post viral injection to match the behavioral retest time-point.

Optogenetic silencing of young abDGCs (at seven week maturation) led to significant BOLD signal changes within the local DG/CA3 regions, as well as in the contralateral hippocampus (*Figure 4C*). No BOLD signal changes were detected in the GFP-control group examined with the same opto-fMRI protocols (*Figure 4B*), confirming that the observed BOLD signals are not due to

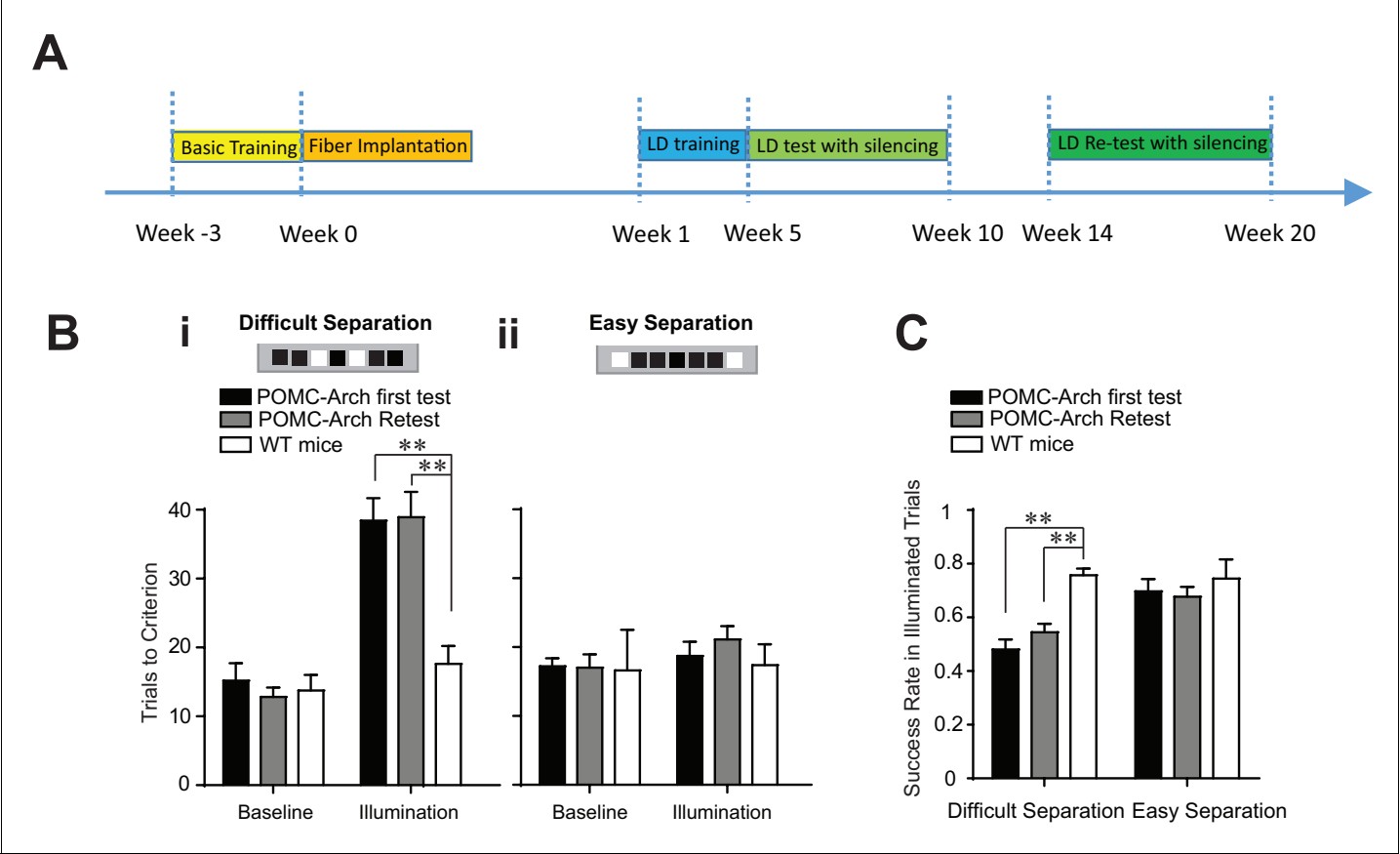

**Figure 3.** Optogenetic silencing of the DG in POMC-Arch transgenic mice impaired behavioral performance in the location discrimination task. (**A**) Experiment timeline (**B**) Behavioral performance measured in the POMC-Arch transgenic group (n = 5) and the WT control group (n = 4), in the baseline blocks and the laser illumination blocks, for the difficult separation condition (**i**) and the easy separation condition (**ii**) [Two-way ANOVA, significant illumination effect: $F_{2,11} = 46.05$, p<0.001; significant group effect: $F_{1,11} = 10.30$, p<0.01; significant mice group × illumination interaction: $F_{2,11} = 6.63$, p<0.05; Post hoc Bonferroni tests for either first test or retest: POMC-Arch vs. control groups for illumination blocks: p<0.01; for baseline blocks: p>0.05, and the easy separation condition, Ps > 0.05]. (**C**) The success rate for trails with laser illumination was significantly decreased in the difficult separation condition in the POMC-Arch group during both the first test and the retest, compared to the control group [Two-way ANOVA, significant illumination effect: $F_{1,11} = 6.82$, p<0.05; significant group effect: $F_{2,11} = 13.39$, p<0.01; Post hoc Bonferroni tests for either first test or retest: POMC-Arch vs. control groups for difficult separation: p<0.01; for easy separation: p>0.05]. **p<0.01. Error bars indicate standard errors of mean (S.E.M.).

any potential heat or light artifact (*Christie et al., 2013*). The time-course extracted from significantly changed voxels revealed a decrease in BOLD signals in bilateral hippocampal regions (*Figure 4Fi, ii*). When the same mice were retested at 16 weeks, optogenetic silencing failed to produce any detectable BOLD signal changes (*Figure 4D*) (*Alme et al., 2010*). The same opto-fMRI testing in the POMC-Arch transgenic mice showed much stronger reduction in BOLD signals in the local ipsilateral hippocampus compared to that observed upon optogenetic silencing of only young abDGCs (*Figure 4E, 4Fiii*), consistent with the theory that a greater number of DG granule cells were silenced in these transgenic mice. Interestingly and surprisingly, we did not observe any BOLD signals in the contralateral hippocampus in the POMC-Arch mice. It is possible; however, that a large recruitment of the DG ipsilaterally overwhelms, and potentially masks, the contrast signal present in the contralateral hemisphere arising from a relatively sparse population of bilaterally influencing young granule cells. Together, these results suggest that abDGCs can engage the bilateral hippocampus during early development. While fMRI cannot directly measure the neural circuits recruited by young abDGCs, anatomical evidence supports the idea that abDGCs may function through the contralaterally projecting hippocampal neurons that are downstream of the optogenetically silenced

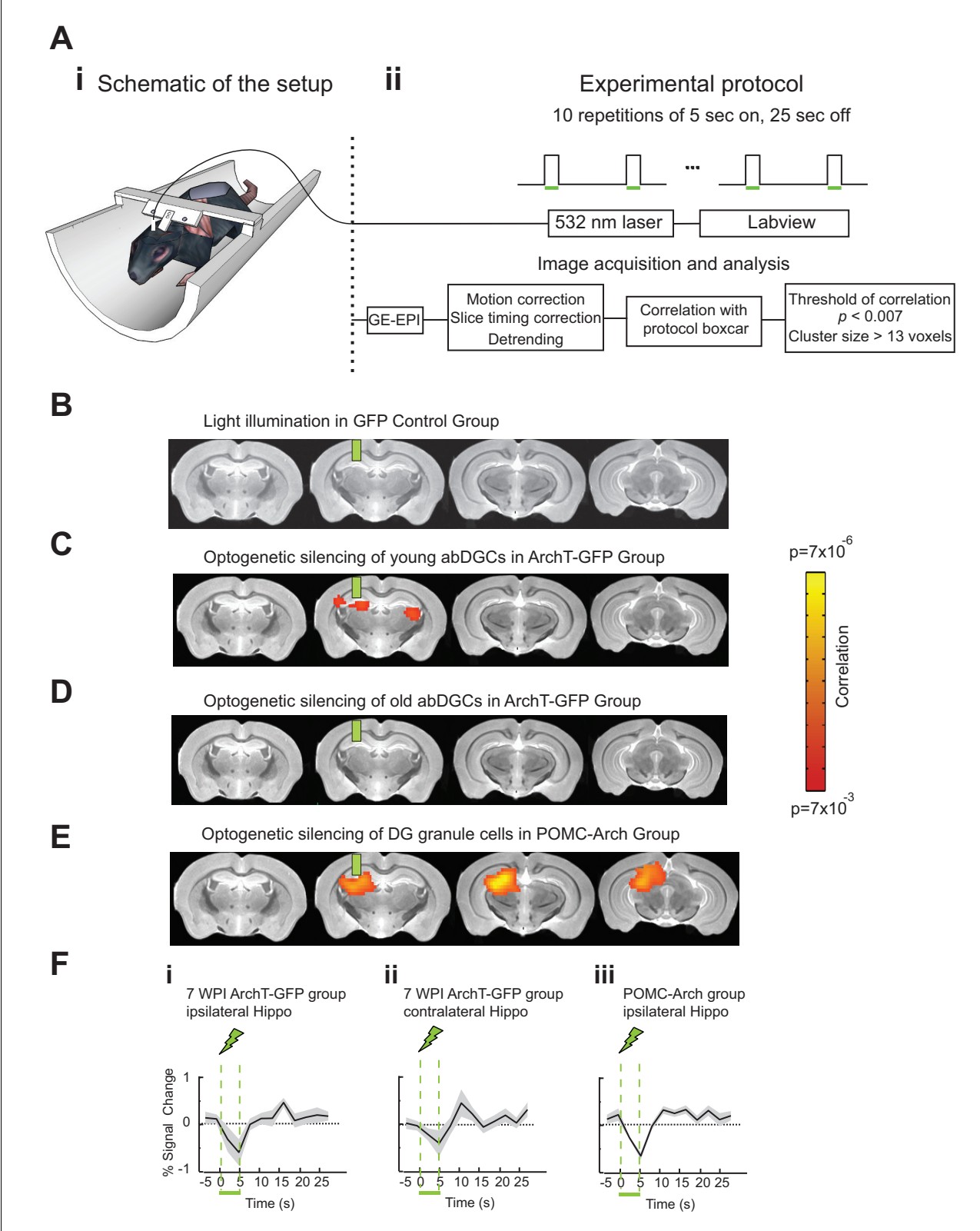

**Figure 4.** AbDGCs effectively modulated BOLD signals in bilateral hippocampus at seven week maturation. (**A**) Opto-fMRI experimental set-up (**i**) and imaging protocol (**ii**). (**B**) No voxels showed changes upon light illumination in the right DG, in GFP-control mice (n = 7 mice). (**C**) Voxels showed significant BOLD signal changes upon optogenetic silencing of young abDGCs (seven week maturation) in the right DG (n = 8 mice). BOLD signals were aligned to the standard mouse atlas. Color indicates the correlation P value for each voxel. (**D**) No voxels showed changes upon optogenetic

*Figure 4 continued on next page*

*Figure 4 continued*

silencing of older abDGCs (16 week maturation, n = 8 mice, same animals as in **C**). (**E**) A lager area is affected by optogenetic silencing of the DG in POMC-Arch transgenic mice, including both developmental-born and adult-born DGCs (n = 5 mice). (**F**) Time courses extracted from the affected voxels of (**i**) ipsilateral and (**ii**) contralateral hippocampus of ArchT-GFP group at 7wpi, and (**iii**) ipsilateral hippocampus of the POMC-Arch group. Green bars indicate the locations of implanted optical fibers.

abDGCs, such as the hilar mossy cells, CA3 pyramidal cells, or CA1 pyramidal cells (*Amaral et al., 2007*; *Toni et al., 2008*).

## Young abDGCs selectively influence the contralateral CA1 region of the hippocampus

To further evaluate the neural circuit influence of young abDGCs in the contralateral hippocampus revealed by fMRI findings, we recorded the local field potentials (LFPs) in the contralateral hippocampus while optogenetically silencing 6–7 weeks old abDGCs (*Figure 5Ai*). 16-channel linear electrode probes were used to record LFPs in awake and head fixed mice injected with retrovirus MSCV-ArchT-GFP or control MSCV-GFP virus (each labeled approximately 200 abDGCs. ArchT-GFP: 198 ± 25.0, GFP-only: 258 ± 56.9, p=0.33, non-paired t-test). A single optical fiber was inserted in the ipsilateral DG to deliver light (5 s every 30 s), and a 16-channel probe was positioned in the contralateral hippocampus and extended into the somatosensory cortex, with the deepest electrode contact positioned in the CA3 region (*Figure 5Aii*). The CA1 region was targeted stereotaxtically at a depth of 950 um below the brain surface, and further confirmed by the appearance of strong theta oscillations during recording, and subsequent histological analysis of probe placement (*Buzsáki et al., 2003*).

During optogenetic silencing of young abDGCs, LFPs recorded in the contralateral CA1 region showed robust increases in power at multiple frequency ranges, suggesting significant neural network influences (*Figure 5D*, middle). In contrast, no change in LFPs was observed in the somatosensory cortex (400 μm above the identified CA1 region), or in the CA3 region (400 μm below the identified CA1 region, *Figure 5D*, left and right). In addition, no change in LFP power was observed upon identical analysis in control mice where young abDGCs were labeled with GFP, confirming that light illumination alone did not produce any detectable LFP change (*Figure 5Ci and 5Cii*). Upon further examination, we discovered that optogenetic silencing of young abDGCs selectively increased higher frequency oscillations, such as beta (15–25 Hz), low gamma (30–50 Hz), and high gamma (70–100 Hz) and high frequency oscillations, but not theta frequencies (4–10 Hz) (*Figure 6A and B*). These results demonstrate that adult neurogenesis is capable of selectively recruiting the contralateral hippocampal CA1 neural network and this occurs without increasing LFP synchrony in CA3. It is important to note that increased activation seen during spatial navigation can promote 30–120 Hz rhythms in CA1 fields without requiring CA3 synchrony at gamma frequencies (*Middleton and McHugh, 2016*).

To further understand this phenomenon, we looked at the relative LFP power temporally across the optogenetic silencing window. We found that the increase in power builds through the silencing period in ArchT-GFP animals [repeated measures ANOVA main effect of time for high frequency: $F_{3,21}$ = 4.367, p=0.015; high gamma: $F_{3,21}$ = 4.426, p=0.015; low gamma: $F_{3,21}$ = 7.230, p=0.002; and beta ranges: $F_{3,21}$ = 5.255, p=0.007] and this effect was not present for any of the frequency ranges in the GFP only expressing animals [all p>0.05]. Further analysis of the time window revealed that the increase in strength was significantly higher during the last 2.5 s of silencing relative to the baseline for all frequencies ranges analyzed. This was followed by a rapid and reversible return to baseline in the 2.5 s period following laser offset, where oscillation power was not different from the baseline period. This gradual increase in power was broadly represented across multiple frequency bands and is suggestive of an increase in generalized excitability possibly via reduced inhibition. In support of this theory, Lacefield et al. also reported that a broad overall increase in gamma power (25–80 Hz) in the hippocampus in urethane anesthetized mice that had been x-ray irradiated 6–10 weeks earlier. Further, they determined this increase was dependent on perforant pathway inputs (2012). Our findings with optogenetic silencing are also consistent with other studies of chronic ablation of adult neurogenesis in that newborn cell silencing altered hippocampal oscillatory patterns

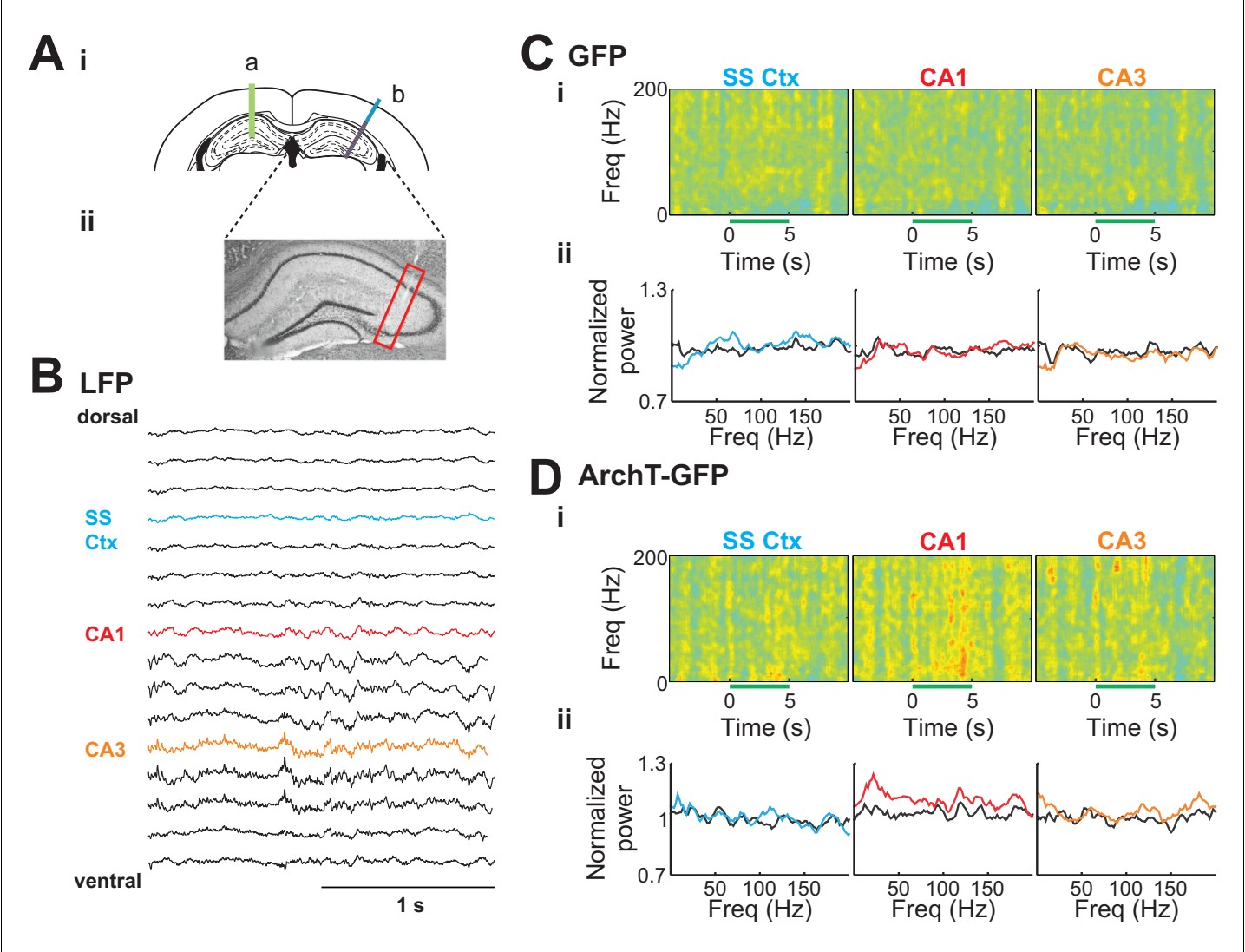

**Figure 5.** Silencing of young abDGCs led to LFP changes in the contralateral CA1 region. (**Ai**) Experimental set up illustrating an optical fiber (green line, a) positioned in the DG where abDGCs were virally labeled, and a 16-channel electrode probe (b) in the contralateral hippocampus at a 30 degree angle. (**Aii**) A representative histological image showing the position of the electrode track in the hippocampus (highlighted with the red box). Cell bodies were labeled with Nissl straining. (**B**) Example LFPs recorded from each of the 16 electrode contacts from the brain surface to the CA3. (**C**) (**D**) Normalized spectrograms of the LFPs recorded in a control MSCV-GFP virus injected mouse (**Ci**) and in a MSCV-ArchT-GFP virus injected mouse (**Di**), and the averaged LFP powers (**Cii**, and **Dii**) before (black trace), and during light stimulation in the cortex (blue), the CA1 (red), and the CA3 (orange).

(*Lacefield et al., 2012*; *Rangel et al., 2013*). Our findings add to this discussion by revealing that this effect occurs immediately and is not a compensatory effect and are supportive of the conclusion that the DG contributes only to a subset of the oscillators within the hippocampus (*Kocsis et al., 1999*). As oscillations represent coordinated interactions of excitation and inhibition in a circuit that may create optimal windows for communication, the ability for abDGCs to alter hippocampal oscillation patterns may be critical for their contribution on behavioral performance (*Buzsáki, 2010*; *Rangel et al., 2013*, *2014*).

## AbDGCs contribution to spatial discrimination depends on learning

We observed that abDGCs can engage contralateral hippocampal neural network and influence spatial discrimination at a time that largely parallels the transient developmental period of elevated membrane excitability and enhanced synaptic plasticity characterized by in vitro studies (*Ge et al.,*

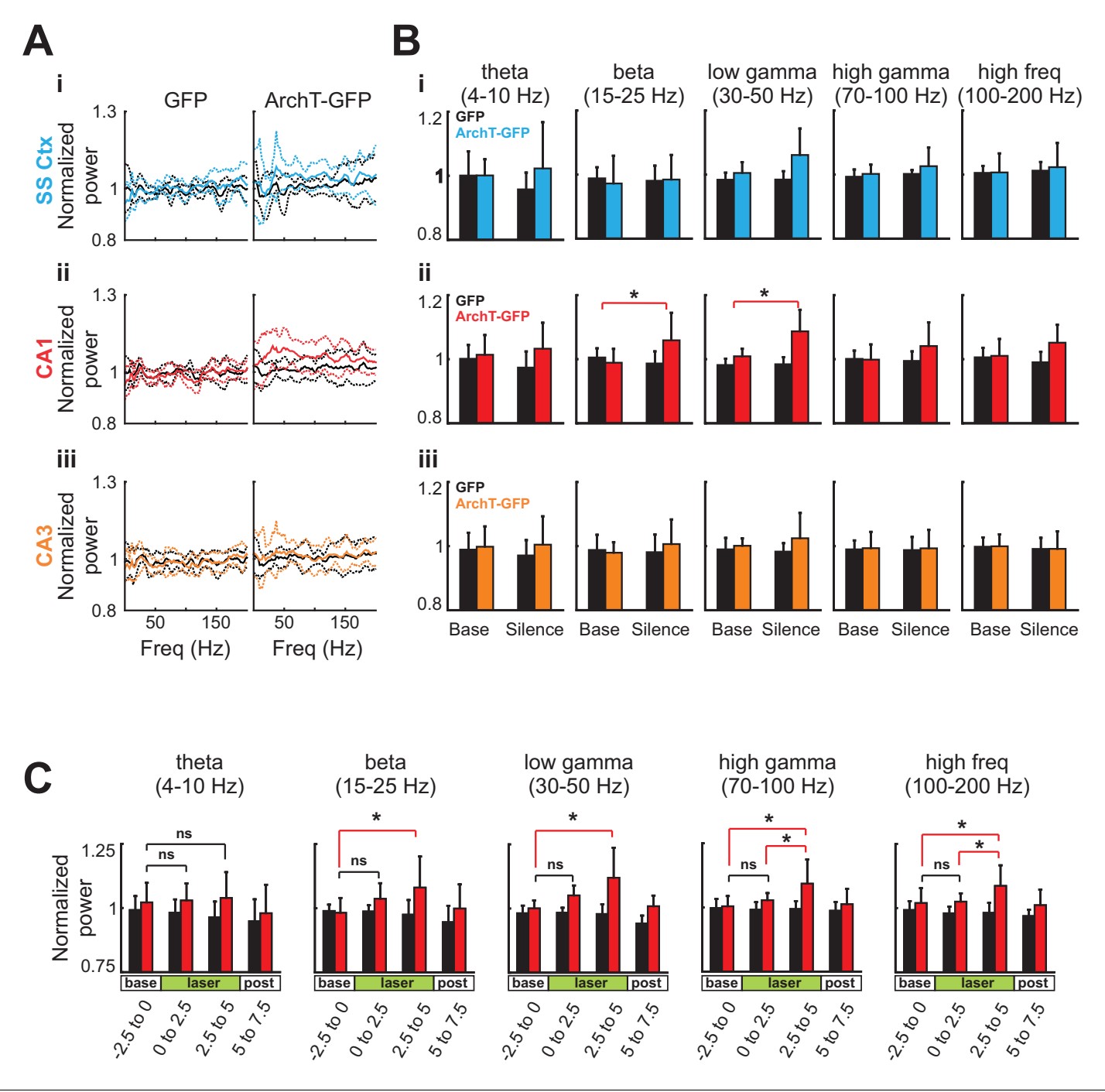

**Figure 6.** Silencing of young abDGCs produces a slow and gradual increase in LFP power across several frequency bands localized to CA1. (**A**) The average normalized power of LFP at different frequencies during the baseline period (black) and during light stimulation in the somatosensory cortex (**Ai**:blue), the CA1 (**Aii**:red), and the CA3 (**Aiii**:orange) in the GFP group (N = 5 mice) and the ArchT-GFP group (N = 8 mice). The solid lines indicate the mean, and the dashed lines are mean ± S.D. (**B**) Averaged LFP powers at various frequency ranges before and during light stimulation. Light stimulation increased LFP power significantly at beta (15–25 Hz) and low gamma (30–50 Hz) ranges, but only in the CA1 region (**Aii**:red). Beta: $t_{1,7}$ = 3.43, p=0.01; low gamma: $t_{1,7}$ = 3.91, p<0.01; high gamma: $t_{1,7}$ = 1.97, p=0.09; high frequency: $t_{1,7}$ = 2.12, p=0.07: (*p<0.05). Error bars are standard deviation (S.D.). (**C**) Averaged LFP power across frequency bands for CA1 before, during, and following optogenetic silencing. Light stimulation produced no significant increases in LFP power in the first 2.5 s of stimulation (ns). Significant increases in LFP power were present in the final 2.5 s of silencing at beta (15–25 Hz) low gamma (30–50 Hz), high gamma, (70–100 Hz), and high frequency (100–200 Hz) ranges: Beta: $t_{1,7}$ = 3.16, p=0.01; low gamma: $t_{1,7}$ = 2.90, p=0.02; high gamma: $t_{1,7}$ = 2.47, p=0.04; and high frequency: $t_{1,7}$ = 2.78, p=0.03: p<0.05, ns = non-significant. Error bars are standard deviation (S.D.).

*2007*; *Gu et al., 2012*; *Dieni et al., 2013*). However, it is unclear how abDGCs biophysical and synaptic properties relate to their neural network contributions and subsequent behavioral influence. Intriguing hypotheses have suggested that young abDGCs may affect overall DG neuronal excitability non-selectively through gross recruitment of hilar inhibitory neural networks as described earlier (*Jung and McNaughton, 1993*; *Ikrar et al., 2013*; *Piatti et al., 2013*), or that young abDGCs may themselves selectively encode information important for behavioral performance through experience dependent tuning of integration into relevant neural networks (*Aimone et al., 2009*). To test whether abDGCs' contribution to the LD behavior is experience-dependent, we examined their role in LD performance after the animal had reached asymptotic performance on the task.

We first completely trained a cohort of mice to perform the location discrimination task, and then injected retrovirus MSCV-ArchT-GFP or control MSCV-GFP to label abDGCs that were naïve to the task. Optogenetic silencing of naive young abDGCs at 5–10 weeks of maturation failed to impair behavioral performance measured either by trial to criteria to complete the illumination block [repeated measures two-way ANOVA, no significant illumination effect: $F_{1,9} = 2.547$, p=0.15; group effect $F_{1,9} = 1.02$, p=0.34; mice group $\times$ illumination interaction: $F_{1,9} = 0.18$, p=0.68], or by success rates for individual laser illumination trials [repeated measures two-way ANOVA, no significant illumination effect: $F_{1,9} = 0.30$, p=0.60; group effect $F_{1,9} = 0.12$, p=0.74; mice group $\times$ illumination interaction: $F_{1,9} = 0.82$, p=0.39] (*Figure 7*). Compared to that in the longitudinal group, retrovirus labeled an identical number of abDGCs in the naïve group, confirming that the lack of a behavioral deficit in the naïve group cannot be explained by the number of labeled abDGCs (*Table 1*).

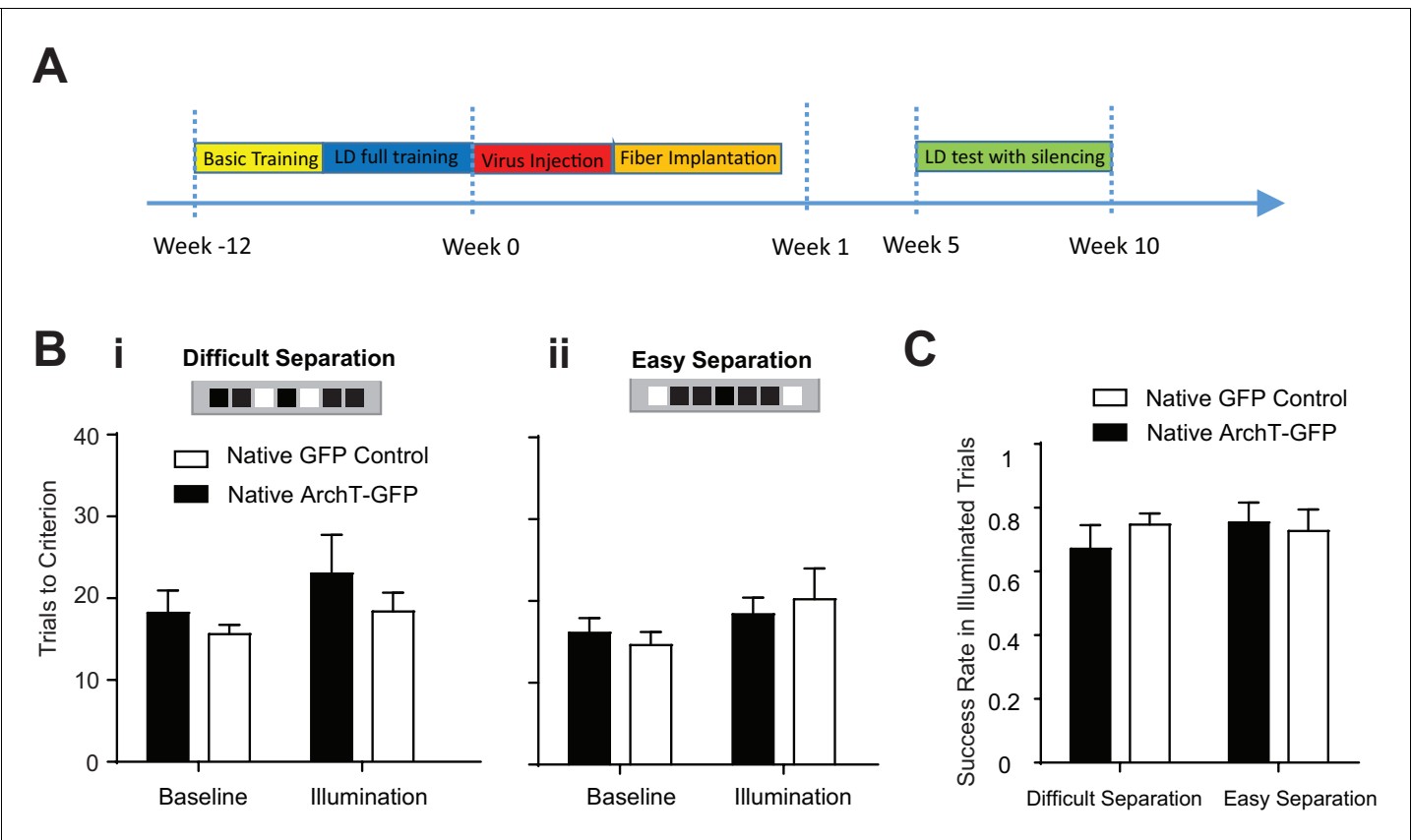

**Figure 7.** Learning influenced abDGCs impact on location discrimination. (**A**) Experimental timeline. (**B**) Behavioral performance measured as trials to criterion in the baseline blocks and the laser illumination blocks, in naïve ArchT-GFP group (n = 6) and naïve GFP-control group (n = 5) performing (**i**) the difficult separation condition, and (**ii**) the easy separation condition. No significant difference was detected between groups [Difficult separation: repeated measure two-way ANOVA, illumination effect: $F_{1,8} = 0.74$, p=0.41; group effect: $F_{1,8} = 0.31$, p=0.60; mice group $\times$ illumination interaction: $F_{1,8} = 3.16$, p=0.12]. (**C**) There are no differences in success rates for naïve ArchT-GFP group and naïve GFP-control group performing either difficult separation or easy separation (Repeated measures two-way ANOVA, p>0.05). Error bars represent standard errors of mean (S.E.M.).

Together, these results suggest that although naïve young abDGCs are capable of altering neural networks detectable by fMRI and electrophysiological measurements (*Figures 4*, *5,* and *6*), their impact on behavioral performance is highly preferential to the networks they are recruited into during the learning experience (*Figure 7*). These findings demonstrate that although abDGCs may play important roles in non-selectively modulating DG neural network excitability, they preferentially influence certain behavioral aspects, likely through experience-dependent shaping during their neural integration.

## Discussion

Here we demonstrate that abDGCs, during an early development period, are highly effective in influencing spatial discrimination and recruiting large scale neural networks involving the bilateral hippocampus. The developmental period identified here, 5–10 weeks after birth, largely parallels the time course when abDGCs exhibit heightened membrane excitability and elevated synaptic plasticity as characterized in vitro (*Ge et al., 2007*; *Marín-Burgin et al., 2012*; *Dieni et al., 2013*; *Kropff et al., 2015*). Our data also parallel other studies in that we find a specialized role for abDGCs in contributing to dentate function, and in particular, young abDGCs, in enhancing DG's pattern separation ability (*Ge et al., 2007*; *Clelland et al., 2009*; *Sahay et al., 2011a*, *2011b*; *Kesner et al., 2014*; *Rangel et al., 2014*).

Since the location discrimination task requires a prolonged training process, it is difficult to further narrow this maturation time window, but these observations highlight how biophysical changes in individual abDGCs may influence their neural circuit integration and subsequent behavioral involvement. Although previous behavioral studies have reported age-dependent influences of abDGCs (*Gu et al., 2012*; *Nakashiba et al., 2012*; *Swan et al., 2014*), our findings add to this discussion by demonstrating that task acquisition coincident with development and maturation is an important contributor in determining if abDGCs will influence spatial location performance. It has been well demonstrated that experience shapes survival, maturation and synapse formation during abDGCs development (*Tashiro et al., 2007*; *Rangel et al., 2014*; *Bergami et al., 2015*). Our study suggests that these considerations also can be applied to neurons recruited into task-relevant networks. The learning dependent behavioral impact further supports the rational that abDGCs have dynamical connectivity and their integration within hippocampal networks is highly plastic, and subject to modulation through interactions with the environment. It is reasonable to conclude that within the observed developmental window, abDGCs biophysical properties ultimately determine their neural network impact through strategic and learning dependent placement of synaptic connections, beyond nonspecific modulation of local networks (e.g., through regulating overall levels of feedback inhibition [*Amaral et al., 2007*; *Myers and Scharfman, 2009*]).

Other studies have indicated that enrichment or behavioral training can influence abDGC survival rate, as well as tune their activity patterns to be more selective and dedicated to familiar experiences. Based on these findings, it is plausible that abDGCs could facilitate pattern separation via experience and learning dependent integration into specific task relevant neural networks (*Tashiro et al., 2007*; *Aimone et al., 2009*; *Rangel et al., 2014*; *Bergami et al., 2015*). Our behavioral data also support this hypothesis, and provided the basis for exploring how these cells might contribute to network dynamics throughout the maturation process. Toward this endeavor, we employed fMRI and electrophysiology techniques to measure network influences across the maturation windows explored in our behavioral studies. It was only when we silenced young abDGCs during the 6–8 week window, but not later on, did we see significant changes in the contralateral hippocampal regions. Electrophysiological analysis of the contralateral hippocampus also revealed broad changes in the power across a number of frequency bands during optogenetic silencing of young abDGCs. Although we saw a general rise in power in CA3, the LFP effects reached significance in CA1, presumably because of the expansive level of divergence from CA3 to CA1. CA3 pyramidal cells make 13 k–28 k synaptic connections with CA1 pyramidal cells, and each CA3 cell contacts between 4–8% of all CA1 pyramidal cells (*Bezaire and Soltesz, 2013*). The increase in dominant 30–200 Hz rhythms in CA1 fields also occurred without CA3 synchrony at those same frequencies (*Middleton and McHugh, 2016*).

It is intriguing that only the young population had the ability to regulate the hippocampus bilaterally based on our electrophysiology and fMRI experiments. The processes by which these cells

demonstrate this time-limited influence is unknown, but likely involve a combination of factors that coincide with maturation, including altered connectivity and decreased excitability. For example, young abDGCs are easier to stimulate in slice electrophysiology studies (*Ge et al., 2007*; *Dieni et al., 2013*) and are believed to exhibit higher firing rates (*McHugh et al., 2007*; *Deng et al., 2010*), whereas the majority of the DG granule cells, the developmentally-born ones, exhibit low baseline firing rates of under 0.5 Hz (*Jung and McNaughton, 1993*). Therefore, silencing the highly active young abDGCs could produce a significant downstream network effect that is not present when silencing low firing older granule cells. Another possibility is that young abDGCs may exhibit unique connectivity patterns, such as preferential connections to interneuron populations. Dentate GCs send the largest number of their inputs to hilar inhibitory neurons (*Acsády et al., 1998*) therefore it is plausible that silencing young active cells could lead to widespread disinhibition of the hippocampus. This interpretation is supported by a follow up study where *Ikrar et al. (2013)* found that enhancing neurogenesis lead to an increase in synaptic connections of newborn neurons specifically onto hilar interneurons. Interestingly, ablation of adult neurogenesis also leads to enhanced feedforward excitation to CA3 from existing mature GCs.

While it has been reported that an increase in high frequency hippocampal power and overall excitability was present in x-ray irradiated mice several weeks after irradiation, the mechanism by which this occurs is not well understood. Lacefield et al. proposed two possible explanations for the paradoxical finding of how the elimination of young, excitable, newborn cells may lead to enhanced spontaneous activity (2012): First, that the loss of young cells that preferentially target hilar interneurons or more effectively drive them are important for suppressing DG network activity more broadly. Secondly, that an ongoing competition exists for perforant path inputs amongst competing granule cells and that young cells effectively out compete older cells for input. However, in the absence of young cells, mature GCs can be more effectively driven by perforant path inputs and that this would develop over time as a compensatory effect. Our findings add insight into these competing hypotheses in that the increases in hippocampal network activity following x-ray irradiation is not likely due to plasticity changes associated with the loss of these neurons. Rather, it is an immediate effect of disinhibition of the hippocampal network. We found that optogenetic silencing produced changes in the LFP power that ramped as silencing persisted with the end of the silencing window showing enhanced excitability when compared to the early window. If mechanisms related to synaptogenesis were solely responsible for these changes, it is unlikely that an optogenetic effect would produce a direct change in excitability over such a short timescale. In support of this argument, *Singer et al. (2011)* described that ablation of neurogenesis also leads to a reduction in inhibitory input into existing granule cells of the DG.

Our data combined with other lines of evidence noted above supports a model where young cells, due to their highly active state, normally contribute to a strong inhibitory tone through hilar neuron connectivity, which is lost during transient optogenetic silencing. The released inhibition leads to a state where unregulated perforant path input promotes an increase in excitability that is conveyed throughout the hippocampal network. Importantly, this result also adds insight into the role that young cells play in inhibiting the hippocampus during periods of pattern separation. During such performance, recruitment of abDGCs would lead to a sparse firing condition within the DG, in which only the most highly trained inputs overcome inhibition prioritizing input specificity in the face of similar competing inputs. (*Jung and McNaughton, 1993*; *Alme et al., 2010*; *Neunuebel and Knierim, 2012*; *Ikrar et al., 2013*; *Piatti et al., 2013*). As pattern separation, in part, is thought to involve the ability to discriminate highly similar inputs into separate and non-overlapping representations (*Kheirbek et al., 2012*), the loss of global inhibition associated with reduced activity of young GCs could contribute to a decline in input specificity necessary for pattern separation (*Sahay et al., 2011a*).

The process by which young abDGCs can uniquely influence contralateral networks in addition to ipsilateral regions is a key question going forward. It is possible that abDGC influence cells with direct connectivity to CA3 pyramidal cells which has recently been proposed (*Temprana et al., 2015*). Under this scenario, silencing young abDGCs may lead to large network changes in the CA3 network that further recruit the contralateral hippocampus through the hippocampal commissure. Adult born DGCs may also show enhanced connectivity to the contralaterally projecting hilar mossy cell that engage the contralateral CA1 (*Christian et al., 2014*). Further studies will be necessary to disambiguate these possibilities.

Taken together, our findings highlight a remarkable and previously undefined role for young adult-born dentate granule cells in influencing spatial pattern discrimination networks. We found that these neurons are recruited into task-relevant networks in an experience-dependent manner, which highlights a role for the importance of ongoing neurogenesis in adulthood. We also found that this cell population specifically, had the ability to recruit and influence large neural networks including the bilateral hippocampus. This provides in part, a mechanistic explanation for how such a small number of cells can so broadly impact behavior performance and offers insight into how the influence of young abDGCs is well suited for mediating cognitive flexibility in the DG circuits (*Burghardt et al., 2012*). These findings may lead to novel perspectives on how adult neurogenesis modulates hippocampal functions and how a reduction in neurogenesis associated with aging or neurological disease can be related to discontinuities in hippocampal networks more broadly.

## Materials and methods

### Virus design and production

A Murine stem cell virus (MSCV) retroviral vector (*Hawley et al., 1994*) with the human *synapsin* promoter (SYN1) was used to express ArchT-GFP or GFP in adult-born dentate granule cells (abDGCs) (*Figure 1A*). Replication-incompetent retrovirus was packaged via triple transfections of pMSCV-ArchT-GFP or pMSCV-GFP retroviral plasmid, the pseudotyping plasmid pMD2.G encoding for VSVg coat protein (http://www.addgene.org/12259/), and the Gag/pol packaging plasmid (http://www.addgene.org/14887/) into 293FT cells (catalog number R70007, Life Technologies, Grand Island, NY), and then purified through ultra-centrifugation as described before (*Han et al., 2009*). 293FT HEK cells are developed by Life Technologies specifically for high titer retrovirus production, not authenticated, and mycoplasma negative. Virus titer was estimated to be about $10^{11}$ using a retrovirus titer quantification kit that detects viral particles (Cell Biolabs, San Diego, CA). Because this assay also measures inactive virus, it is likely that the infectious titer is lower.

### Measures of viral transfection and specificity

All procedures involving animals were in accordance with the National Institutes of Health Guide for the care and use of Laboratory Animals, and approved by the Boston University and Massachusetts Institute of Technology Animal Care and Use Committee. Female C57B6 mice, 7–8 weeks old, were housed with running wheels for two weeks prior to retrovirus injection. After retrovirus injections, histology was performed in mice at 7, 14, 30 and 60 days, and eight months after injection. In another cohort of mice, dividing cell marker EdU (50 mg/kg) was injected at 0, 3 and 6 days after retrovirus injection, and examined seven days post retrovirus injection

### Histology

EdU was visualized in brain slice using the kit according to the product protocol (Click-it EdU Alexa Fluor Imaging kit, Life Technologies, Grand Island, NY). Nuclei were stained with To-Pro–3 (1:2000, Life Technologies, Grand Island, NY). Fluorescence of GFP, EdU, and To-Pro-3 were visualized with an Olympus FV1000 confocal microscope (Olympus, USA).

Quantification of GFP-expressing abDGCs was performed in both the ArchT-GFP and the GFP-control groups. GFP expressing cells were counted throughout the rostral-caudal extent of the DG. Because of the large size of the hippocampus, we counted approximately eight equally distributed sections per DG (one out of every six 40 μm-thick sections) (*Zhao et al., 2006*). The total number of cells expressing GFP was then estimated for the entire DG. The co-localization of EdU in GFP-expressing cells was quantified in a similar fashion. Nissl staining was also performed to identify electrode tracks after electrophysiological experiment.

### fMRI experiments

#### Surgery and implantation

Female C57B6 mice, 4–5 month old, were injected with MSCV-ArchT-GFP virus (n = 8) or MSCV-GFP virus (n = 7) in the right DG area (AP: −2.0, ML: −1.8, DV: −2.0). One 200 μm optical fiber was implanted at the virus injection site, and a customized headpost was attached to the skull to stabilize the head during fMRI scanning as described before (*Desai et al., 2011*). Four-month-old female

Pomc-Arch transgenic mice (n = 5) went through the same implantation surgery without virus injection.

## Experimental setup and protocol

Data were acquired using a 9.4 Tesla horizontal bore magnet with a 20 cm inner diameter (Bruker BioSpin MRI GmbH, Ettlingen, Germany), and experiments were performed as previously described (*Desai et al., 2011*). Animals were lightly anesthetized with ~1% isoflurane, secured in a custom built G-10 fiberglass MRI cradle via the surgically implanted headpost. A custom-built radio frequency transmit-receive surface coil, specifically designed for the mouse brain, was positioned over the head. A 200 μm optical fiber (Thorlabs, Newton, NJ) was then connected to the optical fiber implanted on the skull. The MRI cradle along with the mouse was then positioned in the center of the magnet bore, locked to the stage to prevent motion. Breathing rate (Small Animal Monitoring 1025, SA Instruments, Stony Brook, NY) and end-tidal expired isoflurane (V9004 Capnograph Series, Surgivet, Waukesha, WI) were continuously monitored throughout the experiment.

BOLD signals were acquired using a gradient-echo echo-planar sequence (GE-EPI) in the coronal orientation. BOLD images were collected with a voxel size of 156 μm × 156 μm × 750 μm, and 10 slices were taken across the entire brain. High-resolution T2-weighted anatomical images (58 μm × 58 μm × 750 μm) were acquired using a rapid acquisition process with relaxation enhancement (RARE) sequence in the coronal orientation, after completing functional data acquisition.

The optogenetic illumination protocol contained a baseline of 60 s of darkness followed by 10 cycles of 5 s continuous laser illumination every 30 s. Laser light (~250 mW/mm$^2$ out of the fiber tip, through a single fiber of 200 μM in diameter) was delivered from a 532 nm green laser (Shanghai Laser Corp., Shanghai, China), controlled by TTL pulses generated through a custom labview script (National Instruments, Austin, TX) controlling a USB Data Acquisition Modules (Cole-Parmer, Vernon Hills, IL). For interested readers this function can be found at the following URL: http://www.bu.edu/hanlab/resources/ or downloaded from the GitHub repository https://github.com/xuehanlab/neurogenesis-manuscript. The optical fiber coupled laser was placed outside the magnet room, with a 200 μm optical fiber (~5 m in length) passing through a small duct into the magnet room.

## fMRI data analysis

Statistical maps for gradient-echo echo planar imaging (EPI) functional data were generated using AFNI (*Cox, 1996*; *Nelson et al., 2006*) (NIH, http://afni.nimh.nih.gov/afni, Bethesda, MD) and MAT-LAB (The Mathworks, Natick, MA). EPI functional data were first motion corrected in all three spatial dimensions without spatial smoothing or undistortion to preserve maximal resolution. Every voxel within each slice was time corrected and detrended, following standard EPI data analysis techniques (*Smith et al., 1999*; *Lindquist, 2008*). Percent signal change for each voxel's BOLD signal was computed by subtracting the baseline BOLD signal, which was calculated by averaging all periods with no laser light illumination throughout the entire scan. Two scans were performed in each session, and the percent signal change was averaged for these two scans, but never across sessions.

To determine which voxels have significant changes in BOLD signal, we performed a voxel-wise time series deconvolution analysis using AFNI's 3dDeconvolve command. Response in a region was deemed significant if a cluster of at least 13 contiguous voxels had correlation p-values at an uncorrected threshold of p<0.007. This cluster size of 13 and this uncorrected p-value threshold of 0.007 were objectively chosen via Monte Carlo simulations performed in AFNI using 'AlphaSim' program, to achieve the multiple comparisons corrected type I error of 0.05, appropriate for statistical analysis of individual voxels taken throughout the entire imaging volume (*Forman et al., 1995*; *Jinhu Xiong et al., 1995*). Individual mouse data were then combined for group analysis using AFNI's 3dAllineate command to align EPI dataset to the reference MRI mouse atlas (*MacKenzie-Graham et al., 2006*). Specifically, individual mouse fMRI data were first aligned to reference mouse atlas, and then combined with AFNI's 3dTcat command to identify brain regions where BOLD signals showed significant changes. Manual alignment was also applied to correct mis-alignment from AFNI computation. For visualization, the group statistical map was overlaid on to the reference MRI mouse atlas (*Figure 4B*). Two experimenters independently performed the data analysis, with one being blind to the group allocation, and both confirmed the results.

To further evaluate the observed BOLD signals in the contralateral hippocampus, additional analysis were performed in individual animals. We confirmed that 7 out of 8 mice showed stronger BOLD signal changes in the hippocampus than in the adjacent thalamus regions. In addition, we performed group analysis excluding BOLD signals from the contralateral thalamus, and found that the contralateral hippocampal BOLD signals remained strong, further confirming that the observed BOLD signals were in the contralateral hippocampus.

Time courses were extracted for brain regions with significant BOLD signal changes from the group statistical map. Extracted time courses were first averaged for all voxels within the selected location, and then averaged across 10 repetitions within each session. For group analysis, extracted time courses from individual mouse in each group were averaged.

## Electrophysiology

13 four-month-old C57B6 female mice were used in electrophysiology studies. Mice were injected with retrovirus (ArchT-GFP: n = 8; GFP-only: n = 5) at a single location in the DG area (AP:−2.0 mm, ML: 1.4 mm, DV: −2.0 mm). Six to seven weeks later, mice were recorded while awake and head-fixed, with one optical fiber (200 um in diameter) inserted into the DG (AP:−2.0 mm, ML: 1.4 mm, DV: −1.8 mm) to deliver 532 nm green light from a laser (Shanghai Laser Corp., Shanghai, China). Two 16-channel electrode probes with electrode contacts 100 um apart (700 um$^2$ electrode contact size for optimal LFP recordings, Neuronexus, Ann Arbor MI, U.S.A.) were inserted into hippocampus bilaterally with an angle of 30 degrees toward center (*Figure 5A*). Each recording session consisted of 100 trials of 5 s continuous light illumination every 30 s. A multichannel Plexon recording system was used to perform the LFP recordings (Plexon, Dallas TX, U.S.A.), and data analysis was performed in Matlab. Due to the light induced artifact on the 16 channel probes positioned within the ipsilateral hippocampus, LFP changes during laser light illumination was not analyzed in that hemisphere. Thus only the contralateral hippocampus recording data were reported. Upon completion of LFP recordings, brain tissue was sliced and stained using nissl staining to confirm the electrode location (*Figure 5A*)

## Behavioral apparatus

The behavior apparatus was designed and built based on previously published studies (*Clelland et al., 2009*; *Creer et al., 2010*). The behavior chamber was constructed with black plastic walls, metal frames, and a perforated metal mesh floor 1 cm above a plastic waste tray. An 10-inch infrared touchscreen (Itouch Systems, Beijing, China) mounted on a 10-inch LCD monitor (Pyle Audio, Brooklyn, NY) was positioned at one end of the chamber. A non-transparent plastic mask containing seven windows (2.5 cm × 2.5 cm, equally spaced, 1 cm apart), 5 cm above the floor and 0.5 cm away from the touchscreen, was used to prevent a mouse from accidently triggering the touchscreen with its tail or other body parts other than its nose. A water pump with an infrared detector was located on the other end of the chamber. A white LED strip was positioned along the top of the chamber wall, serving as the timeout light source. A speaker was positioned outside the chamber to deliver sound cues, and a web camera was used to monitor. The experiment progress was controlled by a customized Matlab script through a National Instrument DAQ (National Instrument, Woburn, MA). All functions associated with each stage of training and touch panel control are available here: http://www.bu.edu/hanlab/resources/ or can be downloaded from the GitHub repository https://github.com/xuehanlab/neurogenesis-manuscript.

## Fiber array for light delivery into bilateral DG

We designed a fiber array containing four optical fibers, 200 µm in diameter each, targeting the DG at two locations in each hemisphere (dorsal location, AP: −2.36; ML: ±2.00; DV: −2.00; ventral location, AP: −3.3; ML: ±2.85; DV: −3.5) (*Bernstein and Boyden, 2011*). The four optical fibers converged onto a single ferrule with an inner diameter of 610 µm. A customized plastic cone was used to package and shield the final assembled fiber array. To facilitate stereotaxic implantation, a fiber targeting Bregma was included to guide the implantation depth, but not to deliver light. To prevent light leakage, black dye was added into dental cement when securing the implant during surgery.

During behavioral training and testing, to ensure animal's free movement, the ferrule on the fiber array implant was connected to a coiled optical fiber (Industrial Fiber Optics, Tempe, AZ), through a

rotary joint (Doric Lenses, QC, Canada) to a 532 nm DPSS laser (Shanghai Laser Corp, Shanghai, China). The laser was controlled by a customized Matlab script via a National Instrument DAQ (National Instrument Corp., Austin, TX) and is available on our website: http://www.bu.edu/hanlab/resources/ or can be downloaded from the GitHub repository https://github.com/xuehanlab/neurogenesis-manuscript. Black tape was used to prevent any light leakage throughout the entire light delivery system.

## Behavioral testing and surgery

Adult 7–8 week old female C57BL/6 mice were used for all behavior experiments. Mice were water restricted throughout the behavioral testing, and were kept >85% of their pre-restriction body weight for the duration of the experiment. Behavioral experiments included four main stages, pre-training, surgery, location discrimination training, and optogenetic testing. To estimate sample sizes, a pilot study using two mice was performed. Power analysis was performed based on these results using G*Power 3.1.9.2 (http://www.gpower.hhu.de) utilizing an $\alpha = 0.05$, a power (1-β) value of 0.80, for a MANOVA repeated measure design.

### Pre-training

Throughout pre-training, mice were housed five in a cage with a running wheel (Bio-Serv, Frenchtown, NJ). Pre-training started two days after water restriction, and lasted up to three weeks. Mice went through three training steps.

1. Habituation and water shaping step: On day one, mice were acclimated to the behavioral chamber for 15 min. The water pump automatically delivered a water drop (10 μl) every minute that coincided with a reward tone (1000 Hz, 500 ms long). On day two and three, mice were placed in the behavior chamber for 15 min; however on this day, all seven windows ports were illuminated during the session. The water pump automatically provided a reward every 30 s, co-delivered with a reward tone. However, mice could also receive extra water rewards by touching any of the seven illuminated windows. A touch also resulted in the audible delivery of a touch tone (500 Hz, 500 ms long) that preceded the reward delivery tone.
2. 'Must touch' step: During this step, mice were required to touch any of the seven illuminated windows on the touchscreen to get a water reward. Touch tone and reward tone always accompanied every touch and reward respectively. A successful trial resulted in an inter-trial timer interval (ITI), lasted for 10 s beginning immediately after a rewarded touch occurred. Daily trainings lasted for 20 min. Once mice were reliably receiving 20 rewards per day, the number of illuminated windows decreased to 3, 2 and eventually 1. The illuminated windows were randomly presented, and if mice touched non-illuminated windows, the trial ended with white LED timeout light presented during the 10 s ITI. Mice proceeded to the final step of shaping upon achieving a >70% success rate.
3. 'Must initiate' step: In the final step of pre-task shaping, mice were trained with only one illuminated window randomly presented, the same as the last phase of the 'Must touch' step, but the next trial would not start automatically. Instead, mice had to poke into the water pump head-entry detector to initiate the next trial, illuminating the choice window. A trial initiation tone (1500 Hz, 500 ms long) accompanied the trigger of the head entry detector. Daily training was completed after 60 trials. Once mice achieved a >90% success rate, they underwent surgery for virus injection and fiber placement.

### Surgery: retrovirus injection and fiber array implantation

After pre-training, mice were given free water access for three days before surgery. Animals were randomly assigned into two groups to receive a stereotaxical injection of retrovirus MSCV-ArchT-GFP (n = 7) or MSCV-GFP (n = 5) (1 ~2 μl per location, injected at a speed of 0.1 μl/min) into bilateral DGs, at two locations within each DG (location 1, AP: −2.0; ML: ±1.5; DV: −1.9. location 2, AP: −2.8; ML: ±2.0; DV: −2.0). During the same surgery, a customized fiber array was stereotaxically implanted to target DG bilaterally. Experimenters who performed the behavior testing were blind to animals' group allocation.

## Location discrimination (LD) test training

LD training was started upon recovery from the surgery, ~7 days after surgery. Mice were first retrained in the 'must initiate' step of the Pre-training stage. Once achieving a >90% success rate in 2 out three consecutive days, they proceeded to LD test training. LD training consisted of 60 trials per day with a fixed ITI of 10 s. In the LD condition, two windows were illuminated but only one of the two illuminated windows was the corrected choice, reinforced with water reward. Touching an incorrect window, illuminated or dark, would result white LED timeout light during the ITI. Mice were first trained to perform the LD test on the medium separation condition, defined as being capable of completing at least two blocks per day in two out of three consecutive days. The medium separation condition consisted of 3 dark windows separating the two illuminated ones. A block consisted of a period where the animal returned to the correct port on seven out of eight consecutive trials. In the subsequent block, the correct and incorrect window location reversed although the two illuminated windows remained constant over the entire training session. On the following day, mice were presented with the same medium separation condition, and with the same correct choice window as the last trial on the previous day. The number of trials needed to complete a block, trials to criterion, was used to measure behavioral performance. Once a mouse was able to complete at least two blocks per day in 2 out of 3 consecutive days, it preceded to difficult separation condition, and then the easy separation condition. Consisted with previous studies (*Clelland et al., 2009*; *Oomen et al., 2013*), mice need few trials to criterion to complete the task in easy separation condition than in the difficult separation condition.

Once animals reached criterion across all three separation conditions (medium, difficult, easy), it proceeded to the optogenetic testing stage. Two mice were excluded from this study as they had not reached criterion performance across all three separation conditions by the end of the eighth week after virus injection. This intermediate condition was used as a training condition during task acquisition and testing to encourage the use of spatial pattern separation but optogenetic silencing trials only occurred for the most-difficult and easy conditions of the task. This was done in order to discourage the adoption of alternative strategies once silencing sessions began.

## Optogenetic testing (LD test with laser illumination)

During optogenetic testing, laser light (~40 mW/mm$^2$ out of each fiber tip) was delivered only in the trials of the second block, and the fourth block if a mouse was capable of reaching the fourth block (illumination blocks). In these illumination blocks, laser light was only delivered after mice made two correct choices in the previous three trials.

Two types of laser light delivery experiments were performed. First, laser light was delivered while the mice were making the choice. Light was delivered upon initiation of the trial after a nose poke, and terminated at the time of choosing a window by touching the touchscreen. Alternatively, light was delivered during the last 5 s of ITI (ITI illumination). Each mouse was tested for two days in the selected illumination pattern to counterbalance the initial choice window location.

Optogenetic silencing effect on behavioral performance was analyzed through trials to criterion in both baseline blocks and illumination blocks. Sometimes, a mouse failed to complete even one illumination block due to optogenetic silencing. In these cases, the number of trials that the mouse performed in the unfinished block was used as the trials to criterion for that block, which is an under-estimation of the optogenetic silencing effect on behavioral performance. No mice failed to complete at least one baseline block in a testing day.

At the end of the testing, all mice were rested with free water access until the retest phase described below.

## Optogenetic retest

The same groups of mice (ArchT-GFP group, n = 5; GFP-control group, n = 5) were water restricted again for SD test around 4–6 weeks after the first test. Retest procedures were identical to that performed in the first test.

## Behavioral testing upon optogenetic silencing of naïve abDGCs

In this behavioral test, 7–8 week old female C57B6 mice (ArchT-GFP group, n = 6; GFP-control group, n = 5) were trained and tested in a similar way as the longitudinal groups described above,

except that the retrovirus injection and the fiber array implantation were performed after these mice were fully trained in all LD test conditions. The criterion for mice to receive surgeries is to achieve at least 2 blocks in 2 out of 3 consecutive days in both difficult and easy separation conditions. After surgery, mice were tested with optogenetic silencing between 5–10 weeks after retrovirus injection.

### Silencing in Pomc-Arch transgenic mice

Pomc-Arch mice were generated by crossing Ai35 mice (*Madisen et al., 2012*) and Pomc-Cre mice (*McHugh et al., 2007*). All mice were obtained from Jackson Laboratories (Bar Harbor, ME). Three to five months old Pomc-Arch (n = 5) and wild type (WT) C57B6 Female mice (n = 4) went through similar LD test training and optogenetic testing procedures, but no retrovirus injection was performed in these mice.

## Statistical analysis

Statistical analysis for opto-fMRI data are described in the corresponding methods section. For LD test with optogenetic testing, mice group x illumination was examined using repeated two-way ANOVA, followed with Bonferroni post hoc t-test to examine the difference in trials to criterion, success rate and the reaction time. For electrophysiology data, two tailed paired t-tests were used. For histology data, unpaired t-tests and one way ANOVA were used. Data met the assumptions of all statistic tests and F-test has been performed to confirm the equality of variance between the groups that were statistically compared. All statistical analyses for behavior, fMRI, and histology were performed with Prism 6. Electrophysiology data was analyzed in SPSS version 20.0).

## Acknowledgements

We thank Drs R Rajimehr, and F Rong for technical support on fMRI analysis; BD Allen and JG Bernstein for support in constructing optical fiber arrays, N James for support in electrophysiology experiment, Kim T Le, Matt P Elam, and Kevin Guerra for support in behavioral experiment. We also thank Drs Michael Hasselmo, Howard Eichenbaum, and members of the Han Lab for suggestions on the manuscript.

## Additional information

### Funding

| Funder | Grant reference number | Author |
|---|---|---|
| NIH Office of the Director | 1DP2NS082126 | Xue Han |
| National Institute of Mental Health | 5R00MH085944 | Xue Han |
| NIH Office of the Director | R01-DA028299 | Alan P Jasanoff |
| Defense Advanced Research Projects Agency | W911NF-10-0059 | Alan P Jasanoff |
| Pew Charitable Trusts | | Xue Han |
| American Federation for Aging Research | | Jia-Min Zhuo |
| Alfred P. Sloan Foundation | | Xue Han |
| National Institute of Mental Health | 1R21MH109941 | Xue Han |

The funders had no role in study design, data collection and interpretation, or the decision to submit the work for publication.

### Author contributions

J-MZ, Wrote the paper, Designed all experiments, Carried out all aspects of experiments and collected the data, Assisted with preparing the manuscript; H-aT, Assisted with fMRI experiments and data analysis, Assisted with electrophysiology experiments and data analysis, Assisted with

preparing the manuscript; MD, APJ, Assisted with fMRI experiments and data analysis, Assisted with preparing the manuscript; MEB, Designed the behavioral testing device, Assisted with preparing the manuscript; AIM, NTMR, Assisted with behavioral testing, Assisted with preparing the manuscript; ESB, Assisted with design of the fiber arrays; LMR, HJG, Wrote the paper, Assisted with preparing the manuscript; XH, Wrote the paper, Deigned all experiments, Supervised the project

**Author ORCIDs**
Howard J Gritton, http://orcid.org/0000-0003-3194-3258
Xue Han, http://orcid.org/0000-0003-3896-4609

**Ethics**
Animal experimentation: All procedures involving animals were in accordance with the National Institutes of Health Guide for the care and use of Laboratory Animals, and approved by the Boston University and Massachusetts Institute of Technology Animal Care and Use Committee (protocol #'s: BU: 15-010; MIT: 1115-111-18).

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
