## [Decision Letter]

[Editors’ note: a previous version of this study was rejected after peer review, but the authors submitted for reconsideration. The first decision letter after peer review is shown below.]

Thank you for submitting your work entitled "Young adult born neurons enhance hippocampal dependent performance via influences on bilateral networks" for consideration by *eLife*. Your article has been favorably evaluated by Timothy Behrens as the Senior Editor and three reviewers, one of whom is a member of our Board of Reviewing Editors. Our decision has been reached after consultation between the reviewers. Based on these discussions and the individual reviews below, we regret to inform you that your work will not be considered further for publication in *eLife* (but see option below).

Summary:

This study combines behavioral analysis, fMRI and electrophysiological measures as well as optogenetic manipulation to examine the role of adult-born granule cells (GCs) of the dentate gyrus in pattern separation. The authors expressed Archaerhodopin (ArchT) in adult-born GCs in wild type mice as well as in adult-born and mature GCs in POMC animals to investigate their relative contributions in spatial pattern separation tasks. The authors argue that silencing of adult-born GCs up to an age of 7 weeks after viral injection results in a failure of performing difficult pattern-discrimination tasks but not at an age >12 weeks and draw the conclusion that adult-born GCs are important for spatial pattern discrimination. Moreover, the authors show that adult-born GCs modulate neural activity in the contralateral hippocampus during a specific developmental time window. The authors find that the effects on the activity of the contralateral hippocampus and the time-dependent nature of the behavioral effect are specific to the manipulation of adult-born GCs.

The reviewing editor and the two peer reviewers agreed that the study is 'well designed' and that 'the manuscript is good and makes an important contribution' to the field. However, several major criticisms have been formulated which finally resulted in a rejection of the manuscript:

1) A major concern of the reviewers was the lacking prove that the optogenetic manipulation only influenced adult-born rather than a broad GC population including adult GCs. Comparing the behavioral effect of ArchT-mediated silencing of adult-born GCs in WT mice to the silencing of GCs in POMC mice in the retest phase raised the question whether adult GCs may be involved in pattern separation in the retest phase.

2) The authors use a visual pattern separation task in which the spatial distance between two illuminated windows needs to be identified by the mice. It remained unclear if mice indeed use adult-born GCs to separate spatial locations or whether mice apply different strategies in the easy versus hard condition such as an edge versus a non-edge strategy. One of the reviewers suggested to apply the eight-arm radial arm maze (different separation distance between arms) as additional behavioral test for pattern separation to strengthen/test their main conclusion.

3) ArchT-mediated inhibition of adult-born GCs in the ipsilateral dentate gyrus enhanced the power of gamma oscillations in the contralateral CA1 and dentate gyrus but not in CA3. It remained unclear how this phenomenon may emerge because GCs form strong converging excitatory synapses onto CA3 principal cells.

Since all reviewers ranked the study as interesting providing a potentially important contribution to the filed, we propose the authors to carefully revise their study and to re-submit it to *eLife* at a later time point.

*Reviewer #1:*

The study of Zhuo et al. examines the influence of adult-born GCs of the mouse dentate gyrus (DG) on pattern separation behavior by silencing these cells with Archaerhodopin (ArchT). This light-activated protein has been expressed in adult born GCs using a retrovirus with a synapsin promoter. The applied pattern separation task is well established and has been previously published by Clelland et al. (Science 2009). The behavioral study is well performed and some of the findings are interesting. However, I have several questions and the main criticism is as follows:

1) Silencing adult born GCs during the already learned difficult pattern separation task results in a decline in behavioral performance. If the task is repeated 14-18 weeks after learning (when adult born GCs turned into adult GCs; retest phase), silencing of the same GCs which in the meantime turned into adult cells, had no impact on the difficult pattern separation task. The authors argue that this is consistent with the observation that adult born GCs change their function during the course of the maturation process. However, if the same experiments were performed with PMOC-Cre mice in which adult-born and adult GCs express ArchT, silencing of this neuron population impaired mice in performing the difficult pattern separation task during the test and the retest phase. Thus, adult GCs must play a role in the pattern separation task during the retest phase. The possible role of adults GCs is not well explained in the manuscript.

2) Although I initially liked the link between silencing of adult born GCs and fMRI measurements, now reading the manuscript more carefully, I find this link very much constructed. The link to the behavior and the LFP recordings is weak. The authors find that silencing of adult-born GCs in the ipsilateral DG influences (enhances) the BOLD signal on the contralateral DG. Surprisingly, by using PMOC mice, silencing of large GC populations had not influence on the activity (represented as BOLD signal) in the contralateral DG. This is surprising because I would assume that the neuronal population silenced by ArchT should be larger in the POMC mouse rather than the adult-born GC population (labelled by retroviral expression of ArchT). The authors do not provide explanations on why the adult neurons have no influence on the contralateral DG. One easy explanation could be that they are not active. I would have liked to see more experimental explanations on these presented fMRI findings. The presentation of the data it is very descriptive.

3) The authors perform LFP recordings in the CA1 and CA3 area and cortex (400 µm above CA1; here it would be important to define this cortical area) and find that silencing adult born GCs has no influence on the power of LFP signals recorded in the ipsilateral CA3 but increases the power, in particular, on fast rhythmic activity patterns (gamma), in CA1. Since adult born GCs form direct synaptic connections to CA3 principal cells, this finding is again surprising and would require more experimentally driven explanations.

*Reviewer #2:*

The study presented in the manuscript uses converging behavioral, MRI, and electrophysiological measures to study the contributions of adult born granule cells in the DG using an optogenetic manipulation. The authors studied two different transgenic mouse models (Arch-T and POMC-Arch) to investigate the contributions of developing GC's and GC's more broadly (Arch-T and POMC-Arch, respectively). They also used a longitudinal study design to investigate the role of adult-born GC's at different points in their development. The authors find evidence that adult-born GC's influence spatial discrimination performance and modulate neural activity in the contralateral hippocampus during a specific time window. They find that the effects on the activity of the contralateral hippocampus are specific and the time-dependent nature of the behavioral effect are specific to their manipulation of adult-born GC's. They also tested the influence of GC's born after learning of the task and found that they had no influence on performance. They interpret the results as evidence that adult-born granule cells may influence cognition by influencing bi-lateral hippocampal network activity.

In general, the study is well designed. The authors make use of several methods to investigate the question and have good internal controls. This is a good paper and makes an important contribution. However, I have some reservations regarding the choice of the behavioral task and the analysis of that task with regard to pattern-separation. I can understand the choice of task design from a pragmatic standpoint, but it has limitations. The authors discuss their work largely in terms of pattern-separation, and it is clear how the pattern-separation literature could provide the theoretical framework to motivate the study, however, it is not clear that the behavioral task is well suited to dissociate pattern-separation from other possible alternatives. I've outlined my concerns below.

Concerns:

1) The authors describe using an intermediate condition in their paradigm, but only perform analyses contrasting the easy and hard conditions.

2) In the subsection “Adult neurogenesis contributes to location discrimination (LD) performance” the authors write "Since the difference between the easy and difficult separation is the spatial distance between the two illuminated windows, these results indicate that young abDGCs are specifically involved in discriminating finely separated spatial locations […]" In order to make this claim, the authors need to show that it is unlikely that the mice are using different strategies in the easy vs. hard condition. The authors cite Clelland et al. (2009) when referring to their task, but Clelland used an 8-arm radial maze task. The task more closely resembles the task described in Creer et al. (2010). I'm concerned that if it were 7 windows in a row as is depicted by Figure 1, the mice could be using an edge strategy in the easy condition. It would be much more informative to see if there is also a difference in the middle condition, but I'm concerned that they may use an edge/not edge strategy in that condition as well since 2 out of 3 possible configurations include an edge window. It's not quite clear that the differences in behavioral performance between the easy and hard conditions is really the result of the different spatial separations between the conditions, or if it is the result of a difference in the distribution of edge vs. non-edge trials in the conditions. The case for pattern separation could be made much more strongly if the analysis could compare two non-edge conditions at different separations as is the case with the radial arm task from Clelland 2009.

*Reviewer #3:*

This a very interesting manuscript incorporating a large number of studies. The topic is really important. I have only one suggestion and one question.

1) It is important to explain in detail that you know that your optogenetic treatment only affected new born cells rather than mature dentate gyrus cells or hilar cells.

2) I recommend adding the below listed manuscript to your Introduction and perhaps Discussion.

Kesner RP, Xu H, Sommer T, Wright C, Barrera VR, Fanselow MS. Hippocampus. 2015, The role of postnatal neurogenesis in supporting remote memory and spatial metric processing.

---

## [Author Response]

[Editors’ note: the author responses to the first round of peer review follows.]

*[…] Reviewer #1:*

*The study of Zhuo et al. examines the influence of adult-born GCs of the mouse dentate gyrus (DG) on pattern separation behavior by silencing these cells with Archaerhodopin (ArchT). This light-activated protein has been expressed in adult born GCs using a retrovirus with a synapsin promoter. The applied pattern separation task is well established and has been previously published by Clelland et al. (Science 2009). The behavioral study is well performed and some of the findings are interesting. However, I have several questions and the main criticism is as follows:*

*1) Silencing adult born GCs during the already learned difficult pattern separation task results in a decline in behavioral performance. If the task is repeated 14-18 weeks after learning (when adult born GCs turned into adult GCs; retest phase), silencing of the same GCs which in the meantime turned into adult cells, had no impact on the difficult pattern separation task. The authors argue that this is consistent with the observation that adult born GCs change their function during the course of the maturation process. However, if the same experiments were performed with PMOC-Cre mice in which adult-born and adult GCs express ArchT, silencing of this neuron population impaired mice in performing the difficult pattern separation task during the test and the retest phase. Thus, adult GCs must play a role in the pattern separation task during the retest phase. The possible role of adults GCs is not well explained in the manuscript.*

We believe that this concern primarily arises from an incomplete discussion within the manuscript of the differences between the use of viral and transgenic methods and how their results might be expected to differ. This has been corrected with more clearly written methodology and result interpretation sections that we think will address the reviewers concern:

Briefly, the use of POMC mice in this experiment was designed as a positive control to determine if dentate GCs of any age, including those born during embryogenesis, were required for spatial pattern separation performance. Because, POMC-Cre/ArchT mice represent a transgenic cross created as described in the subsection “Silencing in POMC-Arch transgenic mice” and subsection “Adult neurogenesis contributes to location discrimination (LD) performance”, seventh paragraph, every dentate GC, of every age, will begin expressing ArchT shortly after differentiation and ArchT expression will persist in that cell permanently throughout the animal’s life. Silencing thus reflects the effects of silencing every granule cell, of every age simultaneously (including young and old cells at the time of retesting and recapitulates the effects noted in DG irradiated animals). In the case of POMC mice, the distinction between newborn GCs, young adult GCs, and older GC contributions to performance cannot be resolved and provides the rationale for the study outlined below.

Based on the initial findings from POMC mice, we then tested the temporal role of newly dividing GCs with the specific hypothesis that young adult born cells were the primary neurons responsible for the deficits noted in the POMC mice. In order to do so, we conducted the study using the MSCV virus that drives ArchT expression only in newly dividing cells as described in the subsection “Adult neurogenesis contributes to location discrimination (LD) performance”, third paragraph and is limited to cells undergoing division during the week that follows viral delivery. Therefore, by using the MSCV labeling we could limit ArchT expression to a small window of time to isolate the effects of silencing of GC populations of specific ages in a way not possible in POMC mice. The result mentioned by the reviewer, the effects at 14-18 weeks differ between POMC and MSCV mice are to be expected if new young adult born cells are continually recruited to aid in pattern separation performance as older cells mature. In the POMC population, young adult born cells of every age will be also silenced during the retest phase.

Our interpretation of this finding was (first) that GCs are required for pattern separation in this task and (second) that young adult cells likely are continually recruited and contribute to pattern separation performance throughout the animal’s lifetime while adult cells play a limited role as described in the subsection “Adult neurogenesis contributes to location discrimination (LD) performance”, last paragraph. We have modified the flow of the Results and the Discussion to clarify these points and to better highlight the use of POMC mice as a non-temporally (age) specific indicator of overall dentate GC contributions to task performance.

*2) Although I initially liked the link between silencing of adult born GCs and fMRI measurements, now reading the manuscript more carefully, I find this link very much constructed. The link to the behavior and the LFP recordings is weak. The authors find that silencing of adult-born GCs in the ipsilateral DG influences (enhances) the BOLD signal on the contralateral DG. Surprisingly, by using PMOC mice, silencing of large GC populations had not influence on the activity (represented as BOLD signal) in the contralateral DG. This is surprising because I would assume that the neuronal population silenced by ArchT should be larger in the POMC mouse rather than the adult-born GC population (labelled by retroviral expression of ArchT). The authors do not provide explanations on why the adult neurons have no influence on the contralateral DG. One easy explanation could be that they are not active. I would have liked to see more experimental explanations on these presented fMRI findings. The presentation of the data it is very descriptive.*

We agree that the findings for both population measures (fMRI and electrophysiology) need to be better related to the network aspects that could contribute to behavioral performance. We have expanded both the Results and Discussion sections to address this shortcoming. Our primary objective with the optogenetic silencing in the fMRI and electrophysiology studies was to gain insight into how this small population of cells could profoundly influence behavior. The experiments were not designed to dissociate the details of the circuit differences that exist between young and mature granule cells in task performing animals; however, we do believe our findings reveal novel and important implications for the effects of newborn cells on hippocampal networks and provide evidence that help resolve a long-standing question of how young GCs modulate excitability as described later.

We side with the reviewer in that the most plausible explanation of a lack of signal during older cell silencing in MSCV mice is that this population is largely inactive. This is apparent in the fMRI MSCV data at the late silencing timepoint (16 week old cells; Figure 4) and is supportive of a putatively inactive cell population that, when inhibited, does not have a prominent influence on the network in either hemisphere. The relative inactivity of adult GCs is well established and we have included a more thorough discussion of these findings and this interpretation in the revised manuscript as appropriately requested by the reviewer.

With regards to Figure 4, the shift to the POMC mice data has been rewritten to highlight the differences between the two experimental groups. Because POMC mice express ArchT in every GC of every age, stimulation in POMC mice silences the entire ipsilateral dentate gyrus GC population during light delivery. We initially expected to see large signals across both hemispheres in the POMC mice as the reviewer noted. However, it is important to highlight that fMRI is a contrast measure; the figure shows that activity in the ipsilateral hemisphere is substantially stronger than any influences that are occurring in the contralateral hemisphere. It is possible that the ipsilateral effect overwhelms the downstream contrast signal arising in the contralateral hemisphere from DG silencing. As a result, the signal seen in the MSCV mice where only a relatively sparse population of 5-8 week old cells is silenced could be more effective in demonstrating their unique influences on the contralateral hemisphere as shown in Figure 4. We have now included these points in the revised manuscript.

While the fMRI was insightful in highlighting a specialized role in young GC influencing bilateral networks, it was only one measure and it did not satisfactorily address network implications related to behavioral performance in our minds. The addition of the electrophysiology data was designed to support the fMRI results mentioned above by adding a second independent measure of young cell influence on contralateral hippocampal networks. Importantly it has also added insight into the role that young cells play in inhibiting the hippocampus during periods of pattern separation where input specificity is prioritized during periods of similar competing input. This relationship between behavior and the electrophysiology data has been greatly expanded upon in the discussion of the increase in gamma power in the manuscript.

In addition, the electrophysiology results support previous findings that suggest young dentate GCs target presumptive hilar interneurons. This was only briefly mentioned in the original version but we now expand upon a model where our data compliments previous studies. Namely that young cells, due to their highly active state, normally contribute to strong inhibitory tone through hilar neuron connectivity that is lost during silencing. The released inhibition leads to a state where perforant path input drives hippocampal excitability unchecked as described in the Discussion, fourth paragraph. This is outlined in detail in the response to criticism 3 below but this new analysis and discussion provides a direct mechanistic explanation for the behavioral results as requested by the reviewer. As a final point, because of size and methodology constraints, neither the fMRI or electrophysiology experiments could have been done in performing animals. However, the results from those studies alone reveal an important, previously unpublished, and unique association for newborn cells having bilateral regulatory control of the hippocampus using two independent network measure methodologies. These results are important because it speaks directly to how a small population of neurons can have a profound influence on networks that support the behavioral results and we have emphasized this more directly in the revised manuscript.

*3) The authors perform LFP recordings in the CA1 and CA3 area and cortex (400 µm above CA1; here it would be important to define this cortical area) and find that silencing adult born GCs has no influence on the power of LFP signals recorded in the ipsilateral CA3 but increases the power, in particular, on fast rhythmic activity patterns (γ), in CA1. Since adult born GCs form direct synaptic connections to CA3 principal cells, this finding is again surprising and would require more experimentally driven explanations.*

The overlying cortical region should have been defined and we have updated the figures and text to describe the cortical region that we recorded from throughout the manuscript as primary somatosensory cortex. As a correction to the reviewer’s comment, all electrophysiology results are from recordings made in the contralateral hippocampus with the goal to confirm and understand the novel effects revealed from the contralateral hemisphere in the fMRI experiments. While we attempted to measure effects of silencing in cortex ipsilateral to the fiber, we predictably found that recordings from ipsilateral cortex were too contaminated by photoelectric optical artifact on the metal contact electrodes to allow spectral analysis consistent with our previous findings (Han et al., 2009).

The point of the reviewer regarding direct GC-CA3 is relevant regardless of the hemisphere, but the results in the ipsilateral pathway may be different than effects seen in the contralateral pathway depending on how the effects of silencing are conveyed from ipsilateral to contralateral hippocampus (i.e., hilar, CA3, or CA1 projections). This point is explored in the revised manuscript, but briefly, our results suggest that silencing does not lead to an increase in excitability that manifests as enhanced synchrony at any particular frequency band in contralateral CA3. It is important to note that increased LFP synchrony in CA1 fields at gamma frequencies can occur without concurrent CA3 synchrony at gamma frequencies (Middleton and McHugh, 2016).

Our findings are supportive of those results in that optogenetic stimulation enhanced overall power without impacting synchrony at any particular frequency – no frequency range reached statistical significance – in the contralateral CA3. This presumably reflects an increase in generalized excitability through reduced inhibition and suggests that the effects of optogenetic silencing are conveyed to the contralateral CA3 via hilar or CA3-CA3 connectivity. The increase in power in CA3 is likely heavily pronounced in CA1 because of the expansive level of divergence from CA3 to CA1 as described in the Discussion, third paragraph.

With regards to the second point, while it is true that dentate GCs form strong excitatory CA3 synapses, these neurons also send the largest number of inputs to hilar inhibitory neurons and thus the loss of young cells associated with optical silencing could lead to widespread disinhibition of CA3 (Ascady et al., 1998). In a follow up study, Ikrar et al. (2013) demonstrated that enhancing neurogenesis lead to an increase in synaptic connections of newborn neurons onto hilar interneurons and confirmed that ablation of adult neurogenesis leads to enhanced feedforward excitation to CA3 from existing mature GCs as discussed in the Discussion, fourth paragraph. Our findings are consistent with the interpretation that young adult born cells regulate hippocampal networks by gating excitation through regulation of hilar interneurons. Our findings now connect much of this literature by adding data that reflects a transient and reversible irradiation like lesion that contrasts all these aforementioned studies that have only looked at network effects several weeks after chemical or x-ray ablation. This full discussion is now represented in the fourth, fifth and sixth paragraphs of the revised Discussion. The expanded Discussion now strongly connects disinhibition as a prominent mechanism to the experimental results as requested by the reviewer. Briefly, as pattern separation, in part, is thought to involve the ability to discriminate highly similar inputs into separate and non-overlapping representations (Kheirbek et al., 2012), we believe that the loss of global inhibition associated with reduced activity of young GCs could contribute to the loss of input specificity necessary for pattern separation (Sahay et al., 2011).

*Reviewer #2:*

*[…] In general, the study is well designed. The authors make use of several methods to investigate the question and have good internal controls. This is a good paper and makes an important contribution. However, I have some reservations regarding the choice of the behavioral task and the analysis of that task with regard to pattern-separation. I can understand the choice of task design from a pragmatic standpoint, but it has limitations. The authors discuss their work largely in terms of pattern-separation, and it is clear how the pattern-separation literature could provide the theoretical framework to motivate the study, however, it is not clear that the behavioral task is well suited to dissociate pattern-separation from other possible alternatives. I've outlined my concerns below.*

*Concerns:*

*1) The authors describe using an intermediate condition in their paradigm, but only perform analyses contrasting the easy and hard conditions.*

This intermediate condition was used as a training condition during task acquisition and testing to encourage the use of spatial pattern separation but optogenetic silencing trials only occurred for the most-difficult and easy conditions of the task. There were no intermediate optogentic silencing sessions. This has been clarified in the manuscript and was done for several reasons:

First we wanted to be consistent with other studies using the same paradigm. In the Cleeland manuscript, animals were only tested in the difficult and easy separation conditions, no intermediate condition existed. In the Creer study that followed, trials to criterion were only reported for a subset of probe trials (small and large separation conditions) although intermediate separation conditions were used for training (similar to our design). We followed this same approach by allowing animals to train in all 3 conditions, but only performed optogenetic silencing under the two conditions reported by Creer.

Secondly, we wanted to avoid over silencing by running as few silencing sessions as necessary because over silencing has the potential to encourage the adoption of alternative strategies – one of which could be the use of spatial features such as wall edges or lack thereof as suggested by this reviewer.

Third, we wanted to avoid the potential risk associated with more tethered performance sessions and the potential of loss of a headplate in a performing animal. Because animals had to be tethered during habituation, run in the testing condition over multiple sessions, separated by more than a month, one genuine concern was that additional training sessions could further stress and increase the likelihood of a headplate separation.

We have clarified the manuscript to make this point clear in the subsection “Adult neurogenesis contributes to location discrimination (LD) performance”, third paragraph and subsection “Location discrimination (LD) test training”, last paragraph.

*2) In the subsection “Adult neurogenesis contributes to location discrimination (LD) performance” the authors write "Since the difference between the easy and difficult separation is the spatial distance between the two illuminated windows, these results indicate that young abDGCs are specifically involved in discriminating finely separated spatial locations […]" In order to make this claim, the authors need to show that it is unlikely that the mice are using different strategies in the easy vs. hard condition. The authors cite Clelland et al. (2009) when referring to their task, but Clelland used an 8-arm radial maze task. The task more closely resembles the task described in Creer et al. (2010). I'm concerned that if it were 7 windows in a row as is depicted by Figure 1, the mice could be using an edge strategy in the easy condition. It would be much more informative to see if there is also a difference in the middle condition, but I'm concerned that they may use an edge/not edge strategy in that condition as well since 2 out of 3 possible configurations include an edge window. It's not quite clear that the differences in behavioral performance between the easy and hard conditions is really the result of the different spatial separations between the conditions, or if it is the result of a difference in the distribution of edge vs. non-edge trials in the conditions. The case for pattern separation could be made much more strongly if the analysis could compare two non-edge conditions at different separations as is the case with the radial arm task from Clelland 2009.*

This is an interesting question that we may explore in the future but we do not think it takes away from the studies main conclusion that silencing of young adult-born cells impairs spatial pattern separation. We feel the task employed has extensively been validated as a spatial pattern separation task by the literature and it was one of the primary reasons that we selected it.

As mentioned in the Methods subsection “Behavioural apparatus”, we adopted the previously published Timothy Bussey task that was utilized in both the Creer and Clelland papers. We selected this task based on construct validity as a spatial pattern separation task that had validated outcomes in relation to the 8-arm maze in x-ray irradiated animals.

The primary validation comes from the original Clelland manuscript (2009). Within that manuscript they performed DG irradiation and compared performance of this window task (Clelland; Figure 2) relative to an 8-arm version of the radial arm maze (Clelland; Figure 1). In their experiment, they report that on both task variants, DG silencing resulted in profound deficits in task equivalent ways. Near-arm and near-window locations showed profound deficits where far-arm and far-box locations showed negligible impairment in DG irradiated animals. They concluded that both tasks are valid measures of spatial pattern separation and that dentate GCs contribute preferentially to performance in the difficult condition. Thus we feel the point of validation has largely been addressed by the original publication. Also note that the window results of the Clelland study were replicated in our study (similarity of Figure 2 (Clelland, 2009) to the results in 2Ci and 2Cii of the submitted manuscript).

While it is interesting to consider if these same deficits would be present in an 8-arm version of the task, the practicality of performing tethered experiments prevented us from doing so. We chose the window based discrimination task because it had been published and validated as a spatial pattern separation task and it was conducive for use in a tethered animal where an animal must be attached to an overhead optical fiber. A large apparatus like the radial arm maze would require a motorized shuttle system that tracked animal movement and would allow the animal to travel over large distances unencumbered by attachment to a laser diode. This was not a practical option for these experiments and these points are now noted in the subsection “Adult neurogenesis contributes to location discrimination (LD) performance”, second paragraph.

The concern about adopting an edge strategy for easy versions and perhaps to a lesser degree for intermediate versions is a possibility. On any given day, animals perform all trials with only two illuminated windows. In the case of the easy condition, animals are deciding between two edge windows (simple) or two non-edge windows (intermediate and difficult).

Author response image 1.**DOI:**
http://dx.doi.org/10.7554/eLife.22429.010

Animals are faced with a choice of which illuminated window should they enter based on where they entered on the last trial and were they rewarded. The argument of whether edge locations make the easy condition different is a valid one. It is plausible that pattern separation is more integral to task performance during the intermediate and difficult discrimination conditions as neither have edges adjacent to the illuminated windows. Our experiment adopted the use of internal controls (i.e., trials run in the same animals with and without silencing, silencing sessions in the same animals at 8 and 16 weeks) and yielded only differences in performance for the difficult condition at 8 weeks. If we assume the difficult condition requires spatial pattern separation as previously published, the contributions of edges in simplifying the easy condition is not material to the paper’s main results. In hindsight, the inclusion of an additional window (one to either side of the simple configuration) that would always be blacked out would have been more elegant design and avoided the potential of the easy configuration being different in nature.

However, we have edited the fourth paragraph of the subsection “Adult neurogenesis contributes to location discrimination (LD) performance” noted by the reviewer to more accurately highlight the discrimination of the difficult condition between silenced and non-silenced sessions and also removed reference to the easy condition from that statement. Given this concern of alternative strategies in the easy condition, we hope that the reviewer finds that the new sentence more accurately represents the study findings.

*Reviewer #3:*

*This a very interesting manuscript incorporating a large number of studies. The topic is really important. I have only one suggestion and one question.*

*1) It is important to explain in detail that you know that your optogenetic treatment only affected new born cells rather than mature dentate gyrus cells or hilar cells.*

We have added a more detailed description of how the MSCV construct and the use of POMC mice allow us to drive expression to the granule cell populations we are testing. In addition, we have highlighted how POMC expression allows us to target young and mature granule cells alike and how it differs from retroviral and synapsin directed targeting to newly divided cells. This expanded description can be found in the first and third paragraphs of the subsection “Adult neurogenesis contributes to location discrimination (LD) performance”.

2) I recommend adding the below listed manuscript to your Introduction and perhaps Discussion.

*Kesner RP, Xu H, Sommer T, Wright C, Barrera VR, Fanselow MS. Hippocampus. 2015, The role of postnatal neurogenesis in supporting remote memory and spatial metric processing.*

We have included the reference in the Introduction (third paragraph) and Discussion (first paragraph). We thank the reviewer for bringing this study to our attention.